# Unbiased recording and identification of thymic cellular interactomes using synthetic Notch receptors

Raúl Sánchez-Lanzas[1], Amanda Jiménez-Pompa[1], Elise Smith [2], Nital Sumaria[3], Justin Barclay[1], Foteini Kalampalika [1], Daniel J. Pennington [3], Mirjana Efremova [2] ✉ & Miguel Ganuza [1] ✉

Cellular interactions between thymocytes and other immune and stromal thymic cells play a key role in T cell maturation and homeostasis. Previous efforts delineating the cellular interactomes that support T cell development have mostly relied on imaging techniques, genetic deletion of essential molecular factors and bone marrow chimeras. Here, using synthetic NOTCH receptors we took a direct and unbiased genetic approach to fluorescently label cells in physical contact with CD4+ and CD4+CD8+ thymocytes in vivo in mice. Prospective isolation and transcriptional characterization at single-cell level of the interacting cells exposed the thymic cellular interactome that supports these T cells and how ageing erodes these interactions. Cellular interactors included, among others, dendritic cells, B cells, IL-17+ γδ T cells, fibroblast subsets and thymic epithelial cells. Ligand-receptor pair analyses highlighted signals involved in survival, differentiation and antigen presentation. Our work provides a new means to study thymic cellular interactions.

T cells are essential to the acquisition of adaptive immunity[1]. During T cell development cell-to-cell interactions are critical to the selection of diverse TCR specificities[2]. In particular, thymic immune and stromal cells, including cortical thymic epithelial cells (cTEC), medullary TECs (mTECs), fibroblasts (Fb), endothelial (ECs), and dendritic cells (DCs), play a prominent role in positive and negative selection to produce a functional T-cell repertoire[2–4]. Developmentally, following T cell commitment, CD4−CD8− double negative (DN) thymocytes become CD4+CD8+ double positive (DP) cells and begin to express a rearranged T cell receptor (TCR). DP cells differentiate into CD4+ single positive (SP4) and CD8+ single positive (SP8) cells[5,6]. Thymocytes with a TCR able to recognize MHC-peptide complexes pass positive selection, which mainly occurs in the thymic cortex, and migrate into the medulla to undergo negative selection that depletes autoreactive thymocytes[6]. The cortex and medulla constitute the major anatomical regions in the adult thymus[7].

Previous efforts delineating stromal and immune cellular niche components that support T-cell development in the thymus have been mostly based on histological analyses of thymus preparations[8–10], genetic deletion of essential factors in candidate cell types followed or not by the generation of bone marrow (BM) chimeras[2,11–13], high-resolution imaging techniques[14–16], including timelapse 2-photon and 3-photon imaging that have allowed to identify and quantify cell-cell interactions[17–21], and, more recently, single-cell profiling married with algorithms predicting cell-to-cell interactions based on the presence of respective ligand/receptors in interacting cells (e.g., CellPhoneDB[22,23] and NicheNet[24]) and spatial transcriptomics[25,26]. Particularly, in vivo, the generation of BM chimeric mice[2], such as the transplantation of major histocompatibility complex (MHC)-deficient BM into lethally irradiated MHC-sufficient recipients and the reciprocal transplant, were key to demonstrate the requirement of MHC molecules and antigen-presenting cells (APC) in T cell positive selection[2,27,28]. TCR-transgenic and TCR-retrogenic mice expressing specific TCRs are

[1]Centre for Haemato-Oncology – Barts Cancer Institute, Queen Mary University of London, London, UK. [2]Center for Cancer Genomics and Computational Biology – Barts Cancer Institute, Queen Mary University of London, London, UK. [3]Centre for Immunobiology – Blizzard Institute, Queen Mary University of London, London, UK. ✉e-mail: m.efremova@qmul.ac.uk; m.ganuza@qmul.ac.uk

broadly used and provide key insights in BM chimera studies[2,29–31]. In vitro, the culture of fetal thymus organs, thymocyte-stromal cell 3D co-reaggregates[32–34] and thymic slices[2,31] also enables the study of thymic cellular interactions[2].

However, these approaches are technically limited by either the inability to detect actual cell-to-cell physical interactions in vivo, low-resolution, low-throughput, lack of specificity of some Cre-recombinase (CRE) mouse lines, inability to label and purify niche cells of unknown phenotype and/or insufficient sequencing depth to capture rare cells[25,26,35]. Thus, the implementation of additional unbiased research strategies to unveil direct cell-to-cell interactions in vivo in the thymus is of utmost importance and can complement currently available methods.

To reveal the cellular interactomes that support T cells in the thymus, we engineered an unbiased genetic approach, Yin&Yang mice (YY), to fluorescently label in vivo the cells in physical contact with T cells, allowing their prospective isolation and characterization at the single-cell level. This system is based on the previously described synthetic NOTCH receptors (synNOTCH)[36–40]. We coupled this technology with CellPhoneDB analyses, which we previously developed[22], to identify relevant molecular pathways regulating the biology of interacting cells.

YY mice comprise two complementary cellular components: inducible sender "Yin" cells (i.e., T cells of interest induced to express a membrane-tethered GFP, mGFP) and receiver "Yang" cells (all other thymic cells express anti-GFP synNOTCH, which functions as a GFP receptor[36,37,40]). Briefly, cell-to-cell contact between senders and receivers allows membrane-bound mGFP (expressed by senders) to activate GFP Receptors (expressed by receivers). This activates an orthogonal signaling pathway triggering mCherry expression in the interacting receivers from an inducible reporter gene. Thus, niche cells interacting with sender T cells are fluorescently labeled with mCherry and can be prospectively isolated and characterized using single-cell RNA sequencing (scRNAseq) and flow cytometry to taxonomize the cellular interactomes that support developing thymocytes in vivo.

By means of a commonly used *CD4-Cre* mouse line[41], we generated a pool of sender cells mostly composed by CD4[+] single positive (SP4) sender cells and CD4[+] CD8[+] double positive (DP) senders. CD8[+] single positive (SP8) senders were also induced, albeit at a much smaller proportion. Transcriptional profiling of isolated mCherry[+] niche cells revealed the cellular interactomes that support these T cells. Our work provides an atlas of the cellular interactomes in the thymus based on physical cellular contacts in vivo and a new means to study thymic cellular interactions.

## Results

### The Yin&Yang system allows the detection of cell-to-cell contacts in vitro

Recent models showed that synNOTCH technology can be used to label cellular interactions in vivo in *Drosophila* and mice[38,39]. Yet, this technology has not been optimized and applied to effectively define at the single-cell level the cellular interactomes that support specific cell types of interest in vivo. To tackle this, we created the YY system, which harbors three alleles: (1) YY (Fig. 1A), (2) *CRE* recombinase of interest and (3) an inducible fluorescent reporter under a tetracycline-responsive promoter element (TRE) (Fig. 1B).

We genetically engineered and targeted the *YY* transgene into the *R26* locus via CRISPR-CAS9 technology to render the *R26-YY^{frtSTOPfrt-lox-GFP-REC-lox-mGFP}* (*YY^{YY}*) allele. The unrecombined *YY^{YY}* allele encompasses two cDNAs with mutually exclusive expression: a *GFP-Receptor* (*GFP-REC*) flanked by *loxP* sites; and a membrane-tethered *mGFP* (Figs. 1A and S1A). A *CAG* promoter confers *YY* with ubiquitous expression (Fig. 1A). The *GFP-REC* encodes an a-GFP synNOTCH[36,37,40]. synNOTCHs are synthetic proteins based on NOTCH receptors[36,37,40].

They maintain the central NOTCH-transmembrane-domain (tmbNOTCH)[36,37,40] (Fig. 1A, right panel). The extracellular and intracellular domains have been substituted to orthogonally transduce exogenous synthetic pathways[36,37,40]. In anti-GFP synNOTCHs, the extracellular domain is an anti-GFP single-chain antibody that recognizes GFP. The intracellular domain (synNOTCH-ICD) is a tetracycline activator (tTA) able to transduce synthetic extracellular signals by directly activating genes under TRE promoters[36,37,40] (Fig. 1A). Membrane-bound mGFP can activate GFP-RECs on interacting cells similarly to NOTCH ligands activating endogenous NOTCH receptors[36,37,40]. A MYC-tag in the GFP-REC allows its detection via αMYC-tag antibodies (Fig. 1A). FLIPASE-mediated excision of the *3*SV40pA* transcriptional STOP cassette from the unrecombined *YY^{YY}* allele (Figs. 1Ai, ii and S1A) yielded the *R26-YY^{lox-GFP-Rec-lox-mGFP}* (*YY^{REC}*) allele, leading to ubiquitous GFP-REC expression as shown by immunohistochemistry (IHC) and immunofluorescence (IF) in *YY^{REC/REC}* homozygous Receiver mice (Figs. 1C and S1Bi). Widespread expression of GFP-REC in *YY^{REC/REC}* receiver mice makes every cell into a potential receiver (Yang receiver cell). Importantly, mGFP expression was neither detected in the thymus nor in any analyzed cell or tissue in *YY^{REC/REC}* receiver mice by IHC, IF and flow cytometry (Figs. 1C, D, S1Bii and S2A, B). This demonstrates that the YY system does not carry undesired unspecific sender cells prior to genetic CRE-induction. Lastly, CRE-dependent recombination of the *YY^{REC}* allele excises the floxed *GFP-REC* cassette, generating the *R26-YY^{mGFP}* (*YY^{mGFP}*) allele (Figs. 1Aiii and S1A), which enables mGFP expression. To yield widespread *GFP-REC* cassette excision and produce mGFP[+] senders in tissues from all three germ layers we employed the *E2a-Cre* allele[42], which triggers CRE expression at the blastocyst stage[42]. As expected, *E2a-Cre^{+/Cre} YY^{+/mGFP}* Sender mice (*E2a-YY* Sender) expressed mGFP across every tissue as shown by αGFP-IHC (Figs. 1Ci and S1Bii) and αGFP-IF (Fig. 1Cii), and by flow cytometry in peripheral blood (PB) white blood cells (WBCs) and thymic cellular components (Figs. 1D and S2B). Consistent with known mosaic expression of the *E2a-Cre* transgene, we observed different recombination efficiencies across tissues and mice in *E2a-YY* Senders[43] (Figs. 1C, D and S2B).

Overall, these data demonstrated ubiquitous GFP-REC expression in *YY^{REC/REC}* Receiver mice, and that mGFP is only expressed following CRE activity, allowing specific and controlled generation of mGFP-expressing sender cells (Yin sender cells).

To enable fluorescent labeling of receivers in actual cell-to-cell contact with mGFP[+] senders, we employed a preexisting fluorescent reporter allele regulated by TRE: *Col1a1-tetO-H2BmCherry* (*TRE^{Cherry}*)[44]. Physical contact among sender and receiver cells allows mGFP (in sender cells) to activate GFP-RECs in interacting receiver cells (niche cells). This triggers a conformational change in the GFP-REC allowing the proteolytic cleavage of its transmembrane tmbNOTCH domain (Fig. 2Ai). This frees the intracytoplasmic tTA, which then activates expression of the TRE-fluorescent-reporter (H2B-mCherry) in receivers, fluorescently marking putative niche cells in physical contact with senders (Fig. 2Ai).

To test the ability of sender cells to induce fluorescent labeling of receiver cells we isolated mouse embryonic fibroblasts (MEFs) from *E2a-YY* sender embryos to obtain sender MEFs; and from *YY^{REC-REC} TRE^{Cherry/Cherry}* receiver embryos to generate receiver MEFs (Fig. 2Aii, iii). Monocultures of receiver MEFs exhibited two populations according to mCherry expression: mCherry[NEG] and mCherry[LOW] (Fig. 2Aii). Importantly, mCherry[HIGH] cells were only observed upon co-culture with sender MEFs (Fig. 2Aii–iv). This demonstrates that senders can activate the synNOTCH transcriptional pathway and record cellular interactions via fluorescent-marking of interacting cells, allowing their identification and prospective isolation based on fluorescence-activated cell sorting (FACS). These experiments demonstrate that the Yin&Yang model can record cell-to-cell interactions by activating this TRE-regulated fluorescent reporter allele.

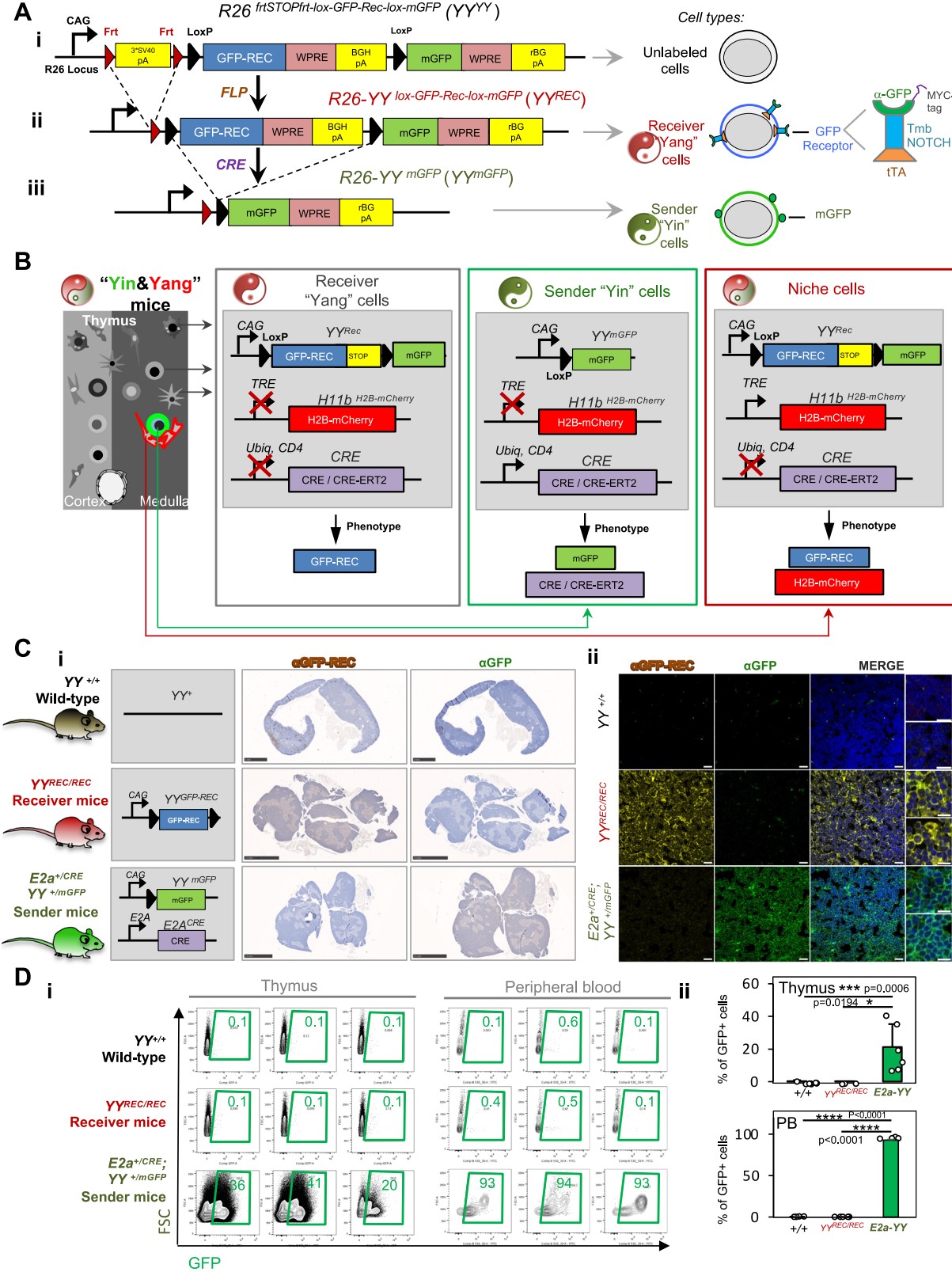

## mCherry-based recording of cell-interactions reports brief, intermittent repetitive contacts (<1 h) and is detectable for at least 6 days and 5 cellular divisions post-encounter

To test the ability of the YY system to record stable and transient cellular interactions, we evaluated the effect of the duration of the interaction on the mean fluorescence intensity (MFI) of H2B-mCherry expression in receiver MEFs (Fig. 2B). Co-culture of sender and receiver

MEFs over time increased both mCherry MFI levels and the number of mCherry^HIGH cells, with MFI reaching a plateau by 39 h post co-culture (Fig. 2B).

To evaluate the ability of the *YY* system to record short interactions, receiver cells were sorted out from senders post co-culture of "sender+receivers" and replated as monocultures for other 36 h to allow mCherry expression (Fig. S3A). As short as a 2-h sender-receiver

**Fig. 1 | The *Yin&Yang* allele allows ubiquitous GFP-Receptor expression and tightly-controlled CRE-dependent induction of GFP sender cells. A** Schematic of the *Yin&Yang* (*YY*) allele. **Ai** The Yin&Yang allele (*R26^frtSTOPfrt-lox-GFP-Rec-lox-mGFP^, YY^YY^*) was targeted into the *Rosa26* locus. *YY^YY^* harbors a CAG-promoter followed by a *frt*-flanked transcriptional STOP cassette (3xSV40 polyadenylation signals, *pA*), a floxed GFP-receptor (GFP-REC) cassette, and membrane-tethered GFP (mGFP). Woodchuck hepatitis virus posttranscriptional regulatory elements (WPRE) increase mRNA stability. Bovine growth hormone (*BGHpA*) and rabbit beta-globin (*rBG pA*) *pAs* ensure transcriptional termination. **Aii** In vivo FLP-mediated excision of 3*SV40pA STOP cassette (via breeding with *R26-FlpO* mice) generated the *R26-YY^lox-GFP-Rec-lox-mGFP^* (*YY^REC^*) allele, resulting in ubiquitous GFP-REC expression producing receiver (Yang) cells. The GFP-REC (a synthetic NOTCH receptor) comprises an extracellular anti-GFP nanobody, MYC-TAG, NOTCH transmembrane domain (TMB) and a cytoplasmic tetracycline transactivator (tTA). Schematic on the right. **Aiii**. CRE-recombinase activity excises the floxed GFP-REC cassette, inducing mGFP expression generating sender (Yin) cells. **B** Alleles and cell types harbored by Yin&Yang mice (+/*CRE; YY^REC/REC^; TRE^Cherry/Cherry^*). Mice carry three alleles: *YY^REC^ TRE^Cherry^*

and a *CRE* driver (constitutive, *CRE*; or tamoxifen-inducible, *CRE-ERT2*). All thymic cells in Yin&Yang mice initially express GFP-REC as receivers. The pattern of CRE expression dictated by the *CRE* allele [*Ubiq-CRE-ERT2 (Ubiq), CD4-CRE (CD4)*] defines sender cells by inducing mGFP. Receiver cells in cell-to-cell contact with senders undergo GFP-REC orthogonal signaling activation and H2B-mCherry expression, fluorescently recording niche interactions (mechanism in Fig. 2A). **C** Representative immunohistochemistry (**Ci**) and immunofluorescence (**Cii**) for mGFP and GFP-REC (detected by αMYC-tag antibody). Images were acquired in three independent experiments from thymuses of *YY^+/+^* wild-type (*n* = 3), *YY^REC/REC^* Receiver (*n* = 3) and *E2a^+/CRE^YY^+/mGFP^* Sender mice (*n* = 2). Data obtained from three histological sections per mouse and experiment. Scale bars: IHC-2.5 mm; IF-20 μm, insets 10 μm. **D** GFP expression in thymuses and peripheral blood (PB). **Di** Representative flow cytometry plots of mGFP⁺ cells. **Dii** Quantification of mGFP⁺ cells in thymus and PB. Data are mean ± SD. Points represent individual mice [*YY^+/+^* (PB: *n* = 4; thymus: *n* = 9); *YY^REC/REC^* (PB: *n* = 5; thymus: *n* = 4); *E2a^+/CRE^ YY^+/mGFP^* (*n* = 6)] acquired over 5 independent experiments. Unpaired two-tailed t-test. ****$p < 0.0001$, ***$p < 0.001$, **$p < 0.01$, *$p < 0.05$.

interaction led to detectable mCherry induction when analyzed 36 h later (Fig. S3Aii, iii). Moreover, three repetitive intermittent interactions of 30 min were enough to yield detectable mCherry induction (Fig. S3B).

Finally, following a 48-h 'senders+receivers' co-culture, mCherry^LOW^ and mCherry^HIGH^ receivers were sorted and plated as mono-cultures to investigate the stability of the mCherry MFI signal with time and cell division (Fig. 2C). H2B-mCherry gets incorporated into the DNA and evenly distributed among daughter cells. Thus, mCherry-MFI approximately halves with each cell division[45,46]. Notably, mCherry-based recording of cell interactions was detectable at least for 6 days post-interaction and 5 cell divisions (Fig. 2C), making it a stable marker of cellular interactions.

Overall, these data demonstrate that the high stability of the recorded signal (i.e., H2BmCherry) enables to record transient cellular interactions, such as those involved in the induction of central tolerance and positive selection, which have been reported to last less than 1 h[17,47].

### The Yin&Yang system enables recording of cellular interactions in the thymus in vivo

The thymuses of *YY^REC/REC^ TRE^Cherry/Cherry^* adult Receiver mice exhibited minimal numbers of mCherry⁺ background cells (Figs. 3 and S3C). Particularly, mCherry^LOW^ cells were very scarce (0.08% ± 0.09%) and mCherry^HIGH^ cells were absent (Fig. 3A, Bi, ii). The presence of very small numbers of mCherry^LOW^ cells was homogeneous across analyzed thymic cell types (Fig. S3C).

To demonstrate the ability of the YY system to detect cell-to-cell interactions in the thymus in vivo, we combined the *YY^REC^ & TRE^Cherry^* alleles with the tamoxifen (TAM)-inducible *Ubiquitin-CRE-ERT2* allele to generate *Ubiquitin^+/CRE-ERT2^ YY^REC/REC^ TRE^Cherry/Cherry^* (*Ubiq-YY*) mice, in which TAM treatment allows CRE-ERT2 to induce the presence of sender cells, thus, that these mice simultaneously have both sender and receiver cells in their thymuses (Figs. 3 and S3D, E). TAM-treatment of 8-week-old *Ubiq-YY* mice yielded ~2.5% of mGFP⁺ senders in their thymuses (2.47% ± 3.12%) (Figs. 3Bi, ii and S3D). Induced mGFP⁺ senders included multiple cell types as Ubiquitin-CRE-ERT2 is non-specifically expressed[48] (Fig. S3D). IF also demonstrated the presence of GFP⁺ sender cells in *Ubiq-YY* thymuses (Fig. 3Biii). The rest of the cells in *Ubiq-YY* thymuses remain as receivers by GFP-REC expression from the *YY^REC^* allele (Figs. 1B, C and 2Ai). The presence of mGFP⁺ senders yielded significant mCherry⁺ receivers in *Ubiq-YY* thymuses (1.31% ± 1.65% mCherry⁺ cells) (Fig. 3Bi, ii), ~15-fold higher than in *CRE-negative* control *YY^REC/REC^ TRE^Cherry/Cherry^* mice (*CRE-negative* Control mice from now on, which carry 0.08% ± 0.09% mCherry⁺ background cells, *p* = 0.0061). Thus, most mCherry⁺ cells in TAM-treated *Ubiq-YY* mice must reflect bona fide cellular interactors with mGFP⁺ senders.

Confocal imaging of TAM-treated *Ubiq-YY* thymuses revealed that mCherry⁺ cells were usually found near mGFP⁺ senders (69.7% ± 24.8% of the times) while only 30.3% ± 24.8% of senders were in physical contact with mCherry⁺ cells (*n* = 62 mGFP⁺ cells, *p*-value < 0.0001) (Figs. 3Biii and S3E). This indicates that detected interactions are typically transient and thymic cells are mobile. Overall, these data demonstrate that the *YY* system can record cellular interactions in the thymus in vivo.

### SP4 and DP cells constitute the major sender populations in CD4-YY mice

To unveil the cellular interactome that supports SP4 cells, we combined the *YY^REC^ & TRE^Cherry^* alleles with *CD4-Cre*[41] (Figs. 4 and 5) and generated the *CD4-Cre^+/Cre^ YY^REC/REC^ TRE^Cherry/Cherry^* (*CD4-YY*) mice. *CD4-YY* mice should carry mostly SP4s, SP8s, and DP sender cells[41], and the rest of the thymic cells should remain as receivers. Cell contact among induced sender cells and the interacting receivers allows activation of the synNOTCH pathway in receiver cells, marking interacting receivers as identifiable mCherry⁺ cells. Flow cytometry analysis of the thymuses in 8-week-old *CD4-YY* mice showed that 7.57% ± 7.44% of the total SP4s were mGFP⁺ (Fig. 4A, Bi). Even though the recombination efficiency for the *YY^REC^* allele was moderate (Fig. 4A, B) (as also observed in *Ubiq-YY* thymuses, Fig. 3Bii), significant numbers of GFP⁺ senders were generated (273,593 ± 336,029 senders per *CD4-YY* thymus, quantified by flow cytometry and cell counts of total thymic cells) (Fig. 4Bii). Most of these senders were SP4s (50%), while 30% were DP and 7% were SP8s (Fig. 4Ci), offering a specific profile of senders more restricted to SP4 cells in 8-week-old mice. The moderate recombination efficiency of the *YY^REC^* allele likely influenced the pattern of labeled cells observed in *CD4-Cre* mice normally encompassing most SP4s, SP8s and DPs[41,49]. Further analysis on the identity of DP senders[31,50–55] revealed that 35% of them were pre-positive selection DPs (CD69⁻TCRαβ^LOW^), 57% were co-receptor reversing DPs (CD69⁺TCRαβ^HIGH^) (Fig. 4Cii). Early post-positive selection DP (CD69⁺TCRαβ^LOW^) senders constituted a minor population (0.6%) (Fig. 4Cii).

Overall, since most of the senders in *CD4-YY* thymuses are SP4s (134,591 ± 136,686 senders) (Fig. 4Bii, C), the majority of mCherry⁺ labeled cells should account for the cellular interactome that supports SP4s. However, given that 30% of the senders were DPs in *CD4-YY* thymuses, ~30% of the detected interactions would be driven by DPs (mainly co-receptor reversing DPs). Likewise, a minor proportion of the identified interactions would be triggered by SP8s (~7%).

### Characterization of the cellular interactome of SP4 and DP cells via flow cytometry and scRNAseq analyses of mCherry⁺ cells

Based on mCherry⁺ recording of cellular interactions in *CD4-YY* mice, we aimed to define the cellular interactomes of SP4 and DP cells

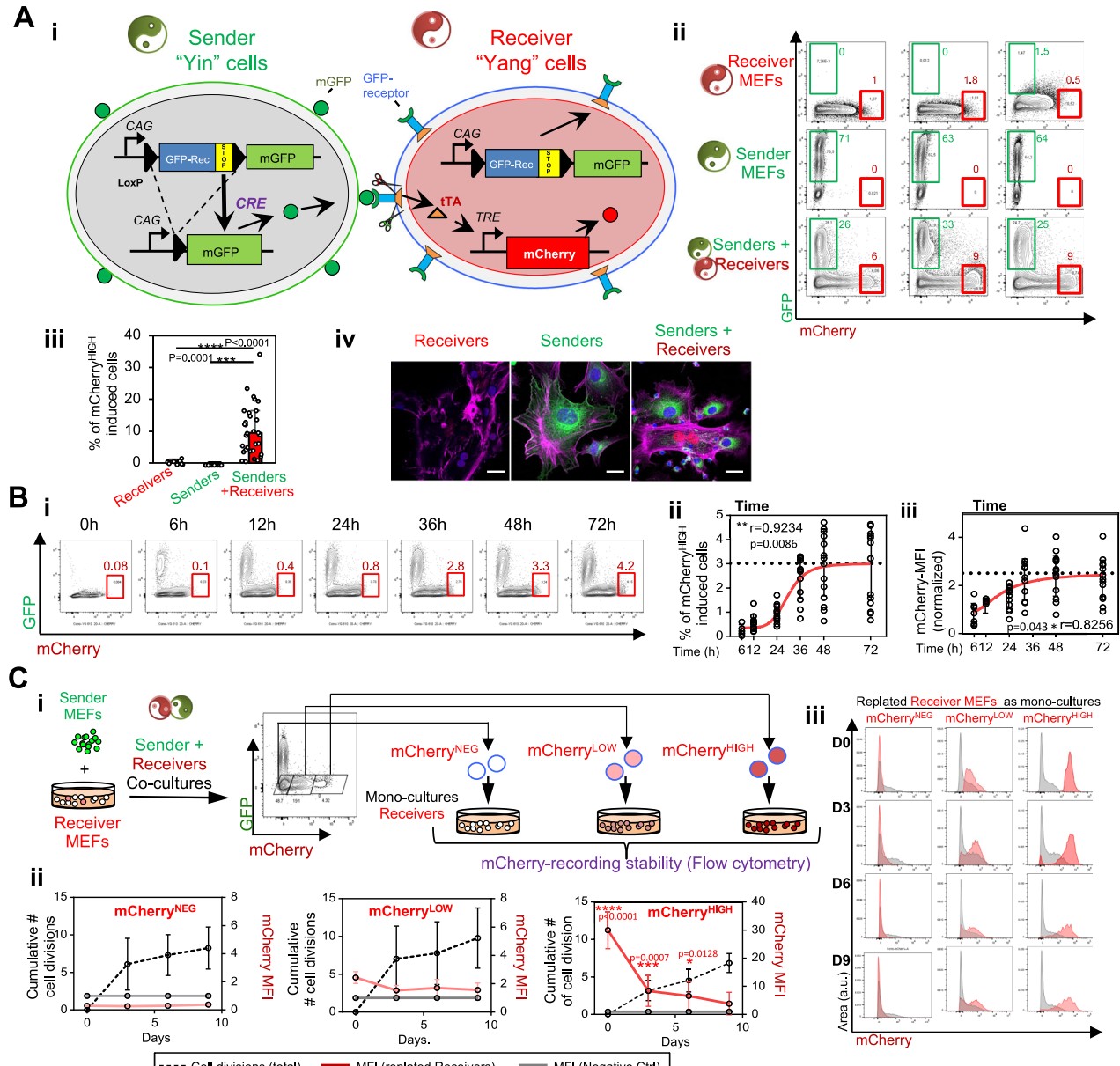

**Fig. 2 | The Yin&Yang system records cell-to-cell interactions in vitro via mCherry labeling. A** Co-culture of sender and receiver mouse embryonic fibroblasts (MEFs) induces an mCherry^HIGH population. **Ai** Schematic of alleles in Sender Yin MEFs (*E2a^{+/CRE} YY^{*/mGFP}*) and Receiver Yang MEFs (*YY^{REC/REC} TRE^{Cherry/Cherry}*). Early E2a-driven CRE activity in the blastocysts of *E2a^{+/CRE} YY^{*/mGFP}* sender mice excises *GFP-REC* cassette, inducing cell-surface mGFP expression. *YY^{REC/REC} TRE^{Cherry/Cherry}* Receiver MEFs express the GFP-REC and carry the inducible H2B-mCherry allele under the TRE operon. Upon sender-receiver contact, mGFP activates the GFP-REC, leading to tTA release into the cytoplasm which can then bind the TRE triggering H2B-mCherry expression from the *TRE-H2BmCherry* allele. **Aii, iii** Flow cytometry of sender-only, receiver-only, and co-cultured MEFs, highlighting induced mCherry^HIGH cells and quantification of mCherry^HIGH cells. [Receivers (*n* = 14); senders (*n* = 10); co-cultures (*n* = 32) from 5 independent experiments]. Unpaired two-tailed t-test. **Aiv** Representative confocal images of co-cultures showing mCherry (red), GFP (green), DAPI (blue), and phalloidin (magenta). Scale bar, 50 μm. **B** Kinetics of mCherry induction following cell contact. Sender and receiver MEFs

were co-cultured and analyzed at 6, 12, 24, 36, 48, and 72 h. **Bi** Representative flow cytometry plots. **Bii** Quantification of mCherry^HIGH cells over time with sigmoidal 4-parameter logistic (S4PL) fit ($R^2 = 0.99$, Df = 2) ($y = 0.3506 + \frac{2.6514}{1+\left(\frac{x}{39.61}\right)^{6.322}}$) (red line). Curve plateauing at 39.2 h. **Biii** Mean fluorescent intensity (MFI) of mCherry over time fitted with S4PL fit ($R^2 = 0.9$, Df = 2) ($y = 0.7872 + \frac{1.7218}{1+\left(\frac{x}{19.56}\right)^{2.314}}$). **Bii, iii** *n* = 15 from three independent experiments. Pearson r correlations (P two tailed) are shown. Dashed lines indicate expected saturation. **C** mCherry-based contact recording is stable over time and cell division. **Ci** After 48-h of co-culture, receiver MEFs were sorted as mCherry^NEG, mCherry^LOW, and mCherry^HIGH, and replated as mono-cultures. **Cii** mCherry MFI over time post-cell contact and across cell divisions (*n* = 4 from three different experiments) $X^2 = 40.71$, Df = 1 ($p < 0.0001$). Uncorrected Fisher's LSD for individual time points. **Ciii** Flow cytometry histograms of sorted receiver MEFs previously co-cultured with senders demonstrate persistence of mCherry labeling for at least 6 days post-contact and 5 cell divisions.

(identified as the major sender populations) by characterizing the mCherry^+ cells in *CD4-YY* thymuses (median age = 7.3 weeks old). mCherry^+ cells were characterized by flow cytometry (Figs. 5A–C and S3F) and scRNAseq (Figs. 6 and S5, 6). In *CD4-YY* thymuses (*n* = 12), flow cytometry analyses revealed the presence of mCherry^+ cells in the SP4/DP-cellular interactomes compared to thymuses in *CRE-negative* control-mice (*n* = 15) (Figs. 5A–C and S3F). These included CD45^+B220^+ cells (containing B cells), CD45^+CD11c^+ cells (containing ~75% of DCs and ~25% of macrophages)[56,57], CD45^+Flt3^+CD4^-CD8^-CD25^- (enriched in DC

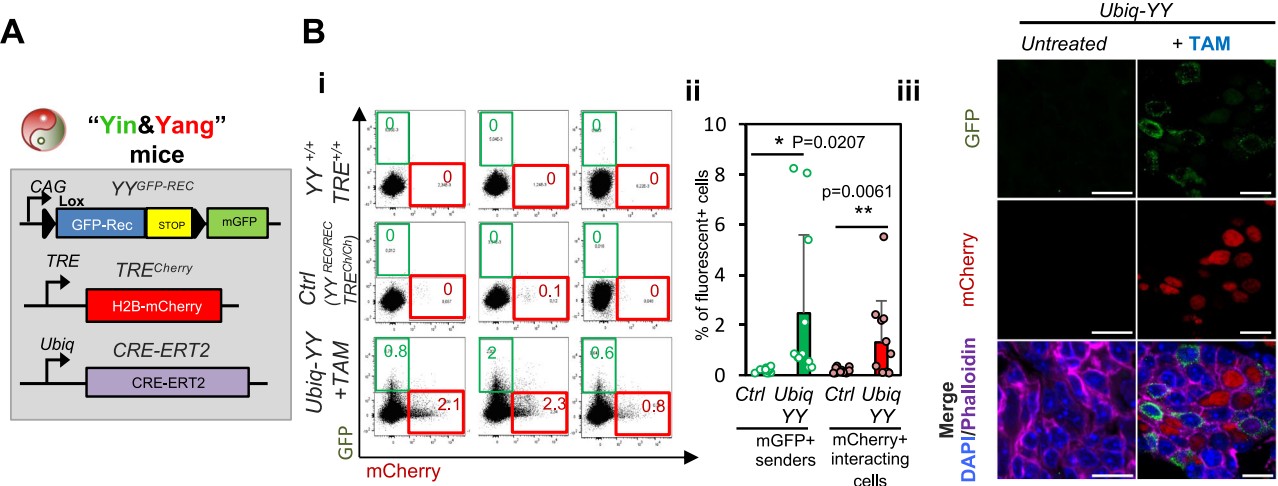

**Fig. 3 | Yin&Yang system allows recording cell-to-cell interactions in vivo in the thymus via mCherry labeling. A** Schematic on alleles harbored by Yin&Yang mice (+/CRE; YY$^{REC/REC}$; TRE$^{Cherry/Cherry}$). **B** TAM treatment (2 mg TAM/day by oral gavage for 11 days) of Ubiquitin$^{+/CRE-ERT2}$ YY$^{REC/REC}$ TRE$^{Cherry/Cherry}$ (Ubiq-YY) led to the presence of mGFP$^+$ sender cells in the thymus which triggered fluorescent labeling of mCherry$^+$ interacting niche cells. Thymuses were analyzed 15 days post-TAM initiation. **Bi** Representative flow cytometry plots of thymuses from TAM-treated Ubiq-YY mice. **Bii** % of fluorescent positive cells (mGFP$^+$ senders and mCherry$^+$ interacting niche cells) for untreated control Ubiq-YY and TAM-treated Ubiq-YY mice. Means and

standard deviations are indicated. Each individual dot represents an individual mouse. Control mice (Ctrl, $n = 16$) and treated Ubiq-YY ($n = 11$), acquired over 9 and 5 independent experiments, respectively. Unpaired two-tailed t-test. **Biii** Representative confocal micrographs of the thymus of TAM-treated Ubiq-YY mice showing mCherry$^+$ receivers in physical and non-physical contact with senders. Scale bars: 10 μm. Quantification in S3D. Means and standard deviations are indicated. Each individual point represents an independent culture. ****$p < 0.0001$, ***$p < 0.001$, **$p < 0.01$, *$p < 0.05$.

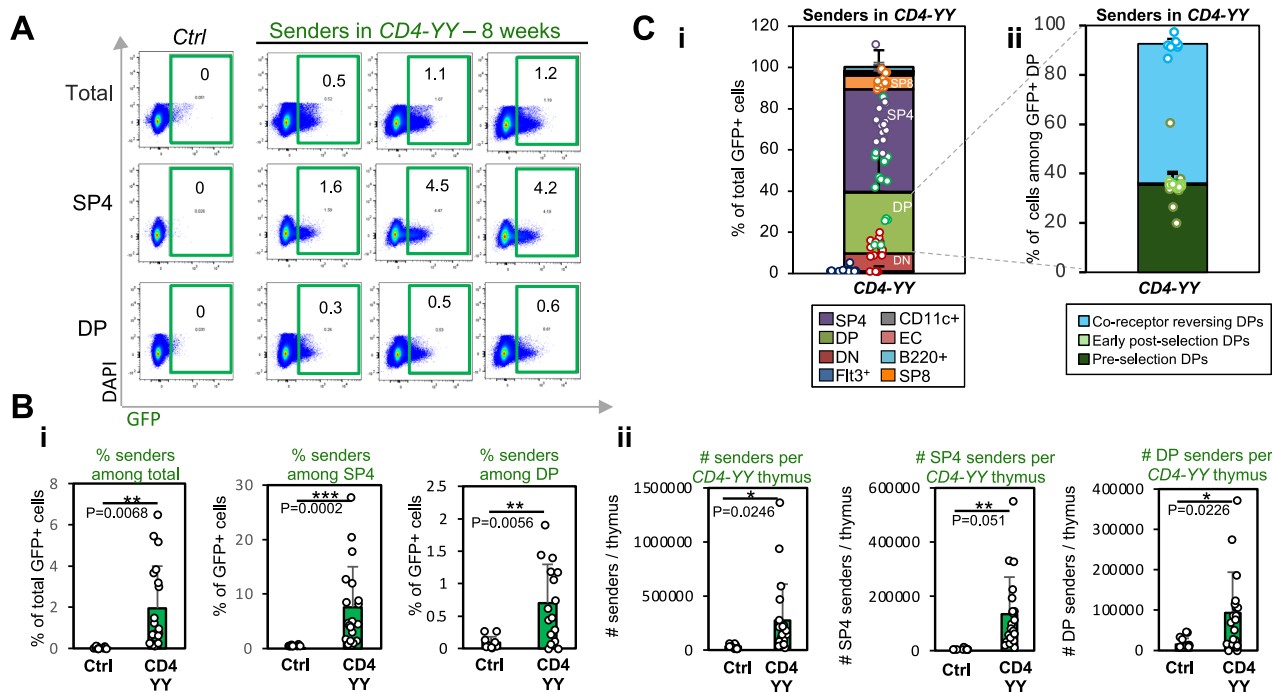

**Fig. 4 | Characterization of the sender populations in CD4-YY mice.** To characterize the sender thymocytes in CD4-Cre$^{+/Cre}$ YY$^{REC/REC}$ TRE$^{Cherry/Cherry}$ (CD4-YY) mice at 8 weeks of age, thymuses from CD4-YY mice ($n = 16$) were analyzed via flow cytometry to determine the presence of mGFP$^+$ senders and their cellular identity. YY$^{REC/REC}$ TRE$^{Cherry/Cherry}$ (CRE-negative) control thymuses (Ctrl) ($n = 10$) were used to detect background. **A** Representative flow cytometry plot showing GFP$^+$ senders among total and within SP4 and DP cells in CD4-YY thymuses. **B** Quantification of total SP4

and DP senders in CD4-YY thymuses ($n = 20$, from 6 independent experiments, unpaired two-tailed t-test). Percentages (**Bi**) and absolute numbers of senders within each compartment (**Bii**) are shown. **C** Cellular identities of GFP$^+$ sender cells in CD4-YY thymuses. **Ci** 50% of senders were SP4s, 30% were DP, 10% DN, and 7% SP8s. 1.9% were B220$^+$ cells and 0.6% CD11c$^+$ cells. All percentages can be found in the Source data file. **Cii** Distribution of DP sender cells ($n = 7$, one experiment). Most DP senders were co-receptor-reversing DPs.

precursors)[58], Lin$^-$cKit$^+$CD44$^+$CD25$^-$ early thymic progenitors (ETP)[59–61], γδ T cells, eosinophils, and endothelial cells) (Figs. 5A–C and S3F). Capsular fibroblasts (CD45$^-$CD31$^-$EPCAM$^-$CD26$^+$)[62–66] were marginally enriched ($p = 0.065$) (Figs. 5A–C and S3F). Most of these cell

interactors (except for ETPs and capsular Fb) concentrate in the medulla. Meanwhile, thymic eosinophils congregate in the cortico-medullary junction and the medulla, and less frequently in the cortex and interlobular septa[67,68]. Interestingly, recent reports indicate that

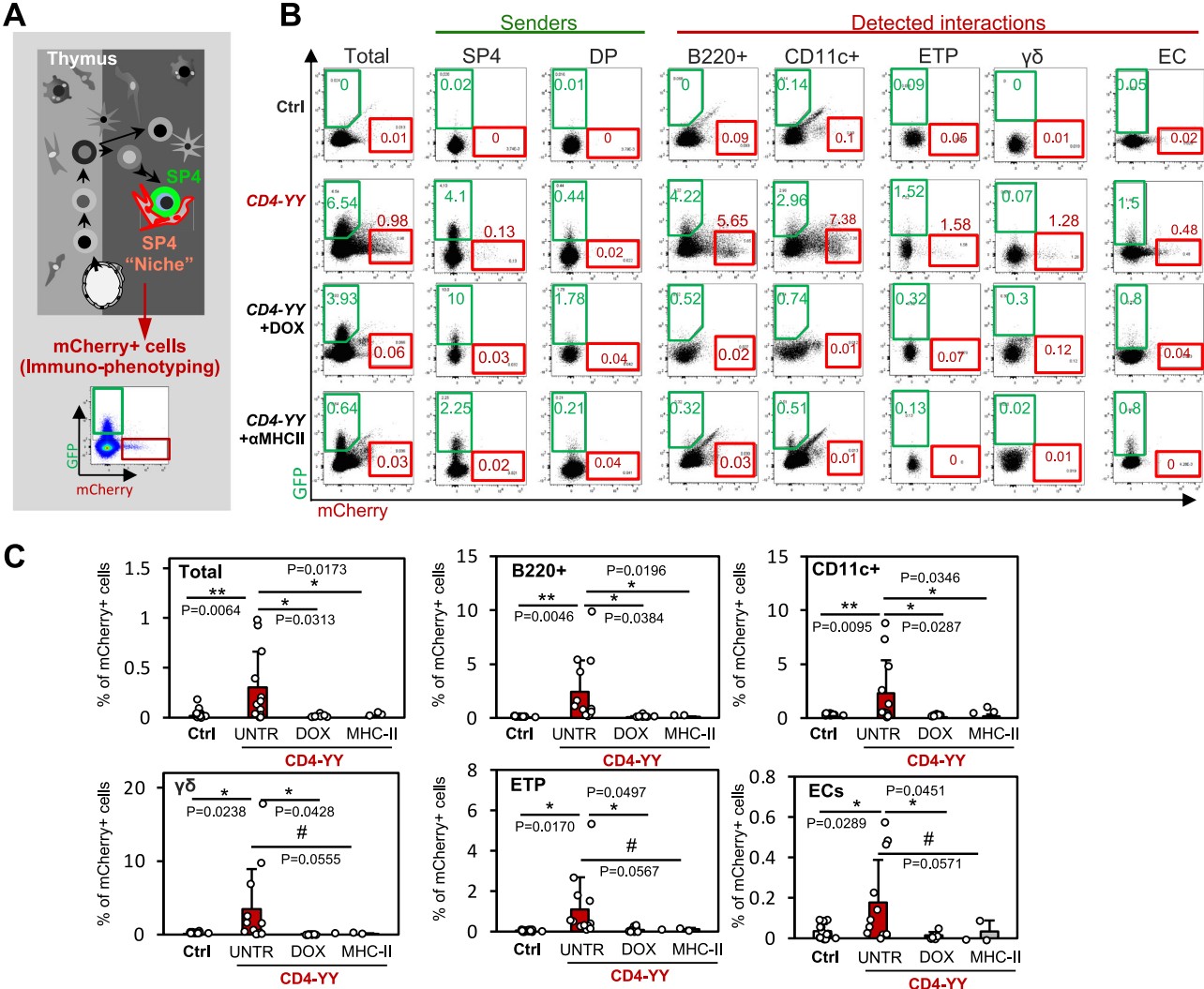

**Fig. 5 | Characterization of the cellular interactome that supports SP4 and DP cells by flow cytometry in young adult mice.** To characterize the cellular interactome (detected as mCherry⁺ cells) that supports SP4 and DPs, thymuses from 6 to 8-week-old *CD4-YY* mice (*n* = 12), and from *YYREC/REC TRECherry/Cherry* CRE-negative control mice (Ctrl) (*n* = 15, from 6 independent experiments) were analyzed for mCherry⁺ enrichment across thymic populations by flow cytometry. To validate the functional relevance of the interactions, 4-week-old *CD4-YY* mice (*n* = 8; four independent experiments) were treated with doxycycline (DOX) for 3 weeks to block tTA activity or with αMHC-II blocking antibody (*n* = 3; two independent experiments) to hinder interactions dependent on TCR – MHC-II. **A** Experimental schematic. **B** Representative flow cytometry plots of mCherry⁺ and GFP⁺ cells within thymic populations. Senders highlighted in green boxes. mCherry⁺ populations in red boxes. **C** Percentage of mCherry⁺ cells was evaluated within each cell population for *CD4-YY* (untreated, UNTR) thymuses and compared to those in Ctrl and treated *CD4-YY* mice (DOX, αMHC-II). mCherry⁺ enrichment in *CD4-YY* compared to controls indicates a cell-to-cell interaction. For instance, -2% of B cells in *CD4-YY* thymuses were mCherry⁺ and, thus, interacting with SP4/DP. Cell populations exhibiting a significant mCherry⁺ enrichment are shown. Note how treatment with αMHC-II blocking antibody and doxycycline erased mCherry⁺ interactions. Frequencies for other analyzed cell populations in Fig. S3F. Means and standard deviations are depicted. Unpaired two-tailed t-test. ****$p$ < 0.0001, ***$p$ < 0.001, **$p$ < 0.01, *$p$ < 0.05, # $p$ < 0.1.

---

eosinophils are essential in thymus recovery from ablative therapy[69] and that some thymic eosinophilic populations bias thymocyte maturation towards SP4 cells[67]. The role of the endothelial cells as part of the blood vessels in T cell development is less understood[3].

Supporting the biological relevance of the identified cellular-interactomes, treatment of *CD4-YY* mice with an αMHC-II blocking antibody (administered from 3 to 6 weeks of age) eliminated most mCherry⁺ cells from *CD4-YY* thymuses (Figs. 5B, C and S3F). These cells included those engaged in well-established interactions between SP4/DPs and APCs, such as DCs and B cells mediated by the MHC-II – TCR molecular axis[3,70,71]. Intriguingly, interactions involving other cell types within the SP4/DP interactome that exhibit low or negligible MHC-II expression[72,73], including SP4, SP8, ISP, DP[74], were also reduced following αMHC-II antibody treatment, which was statistically significant for ISPs but not SP4, SP8, and DP cells (Fig. S3E).

Even though the effect of the αMHC-II blocking antibody on interactions with non-expressing MHC-II cells was, as expected, more subtle, the general reduction in interacting cells suggests that αMHC-II treatment may induce broader perturbations beyond the direct blockade of cognate MHC-II–TCR interactions. Such perturbations may include impaired positive selection of SP4 thymocytes, which could, in turn, disrupt the thymic microenvironment and lead to a reduction in downstream cellular subsets.

These findings align with the well-established interdependence among multiple thymic cell populations required to maintain thymic homeostasis[75,76]. For example, the maintenance of both cTECs and mTECs in adulthood depends on the presence of functional thymocytes[75,77]. Indeed, "thymuses in which T-cell development is arrested at the DN1 stage exhibit a complete loss of cTECs and mTECs"[77], a phenotype that can be rescued by the introduction of functional T-cell

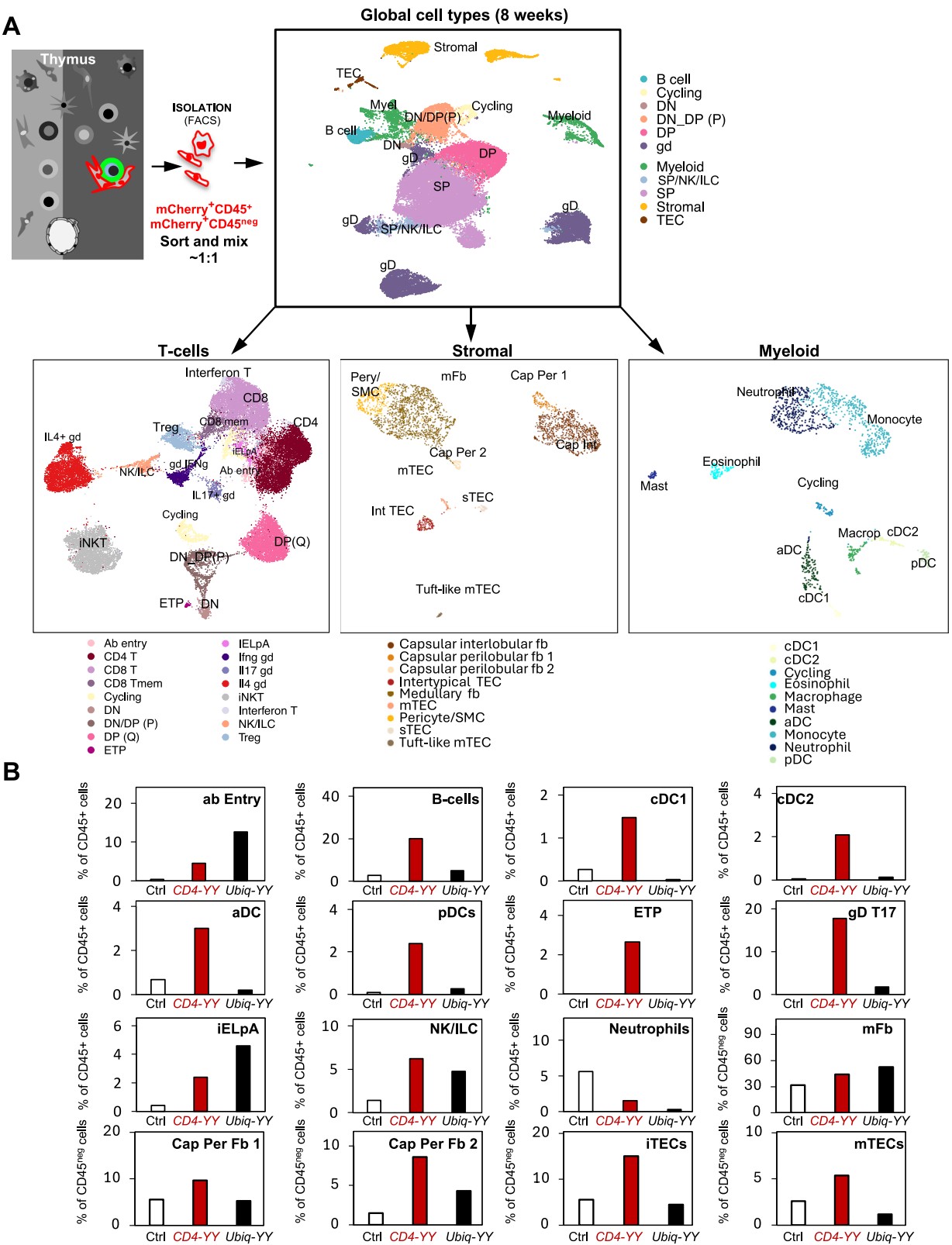

**Fig. 6 | Taxonomizing the cellular interactome that supports SP4 cells by scRNAseq in young adult mice. A** Experimental schematic. Thymuses from 8 to 9-week-old *CD4-YY* mice (pooled from *n* = 8 thymuses), TAM-treated *Ubiq-YY* mice (*n* = 13) and untreated Ctrl mice (*n* = 12) were isolated, dissociated, and pooled for each mouse strain in one experiment. mCherry$^+$ CD45$^+$ and mCherry$^+$ CD45$^-$ niche cells were isolated by FACS. Equal amounts of CD45$^+$ mCherry$^+$ and CD45$^-$ mCherry$^+$ cells were mixed (~14,000 cells each) and sequenced (scRNAseq). UMAP representation of annotated cell clusters from all experiments is shown. UMAP representation for each mouse strain in Fig. S6A. **B** Quantification and characterization of the cellular interactomes of SP4/DP cells. The frequency of each cell cluster was calculated within mCherry$^+$CD45$^+$ and mCherry$^+$CD45$^-$ cells. Cell frequencies for relevant cell clusters are shown. Cell clusters whose cell frequencies in a mouse strain are enriched compared to Ctrl thymuses denote a cell-to-cell interaction. Neutrophils are shown as an example of a non-enriched cluster. mCherry$^+$ enrichment in the rest of cell clusters in Fig. S6B.

progenitors[78]. This demonstrates that, similar to thymocytes, TECs undergo continuous turnover[79]. Conversely, a deficiency of mature TECs profoundly reduces overall thymic cellularity, thereby preventing the development of αβ T cells and invariant NK cells[80].

Thus, it is unsurprising that global MHC-II blockade exerts widespread effects on thymic cellular interactions. More importantly, these results highlight the ability of our system to capture diverse types of cellular interactions, revealing both established and previously unrecognized connections that may open new avenues of investigation.

The transcriptional activity of tTA can be exogenously hindered by doxycycline (DOXY) administration[81] which blocks its binding to TRE, thus preventing tTA-driven TRE-H2BmCherry expression. Hence, to further demonstrate the specificity of the identified interactions, we treated CD4-YY mice (starting at 3 weeks of age) with DOXY. DOXY-treatment completely erased the presence of mCherry+ cells from CD4-YY thymuses in 6–8 week-old mice (Figs. 5B, C and S3F), evidencing the dynamic nature of the identified interactions.

As flow cytometry-based phenotypic identification of thymic populations is constrained by the limited number of employed markers, we transcriptionally characterized mCherry+ recorded interactions via scRNAseq to most efficiently define thymic T-cell interactomes and capture additional cell types. mCherry+ cells were FACS-isolated from the thymuses of CD4-YY (pooled from n = 8 thymuses) and CRE-negative control thymuses (pooled from n = 12, to determine the distribution of background mCherry+ cells) (Figs. 6, S5 and S6A, B). Additionally, mCherry+ cells were also isolated and profiled from TAM-treated Ubiq-YY thymuses (pooled from n = 13), which harbor a much more modest % of SP4-senders (i.e., 20.52 ± 10.82%) (Fig. S3D) compared to the 50% of SP4-senders in CD4-YY thymuses (Fig. 4C) and expected to render differences in the mCherry+ labeled interactomes in Ubiq-YY and CD4-YY thymuses.

We integrated data from all mice and after quality control recovered 41,412 single-cell transcriptomes (Figs. 6, S5 and S6A, B). T cells, including DN, DP, SP8, SP4, Treg, and γδ T cells were the most abundant cells in the dataset (Figs. 6, S5 and S6A, B and Supplementary Table 1). We also identified other immune cells, including myeloid, dendritic, B cells, mast, and natural killer (NK)/innate lymphoid cells (ILC), and non-immune cells in the thymic microenvironment that separated into TECs and stromal cells (Figs. 6, S5 and S6A, B and Supplementary Table 1).

Reanalysis of each individual global cell type exposed diverse cell subtypes. The lymphocytes separated into ETPs marked by expression of Cd34 and Kit, quiescent and proliferating DN (Il2ra, Ptcra) and DP (Cd8+Cd4+) subsets, a transitioning Ccr9+ Tαβ (entry) stage, SP8, SP4, and Tregs, as described before[82]. In addition, we detected a subpopulation of intraepithelial lymphocyte precursor type A cells (iELpA), similar to human CD8+αα T cells, sharing markers such as Pdcd1, Hivep3, and Nr4a3[83]. γδ T cells were subdivided in four clusters: Zbtb16+Il4+γδ, Rorc+Il23r+Cxcr6+γδ, Ifng+Eomes+γδ and T17 γδ with high expression of Sox13, Blk (Figs. 6, S5 and S6A, B and Supplementary Table 1). The myeloid compartment comprised monocytes, macrophages, neutrophils, eosinophils, mast cells, and diverse subsets of DC, including conventional DC, cDC1 (Clecl9a, Xcr1) and cDC2 (Cd1c, Clecl10a), Ccr7+Lamp3+ activated DCs (aDCs), and Lilra4+ plasmacytoid DCs (pDCs) (Figs. 6, S5 and S6A, B and Supplementary Table 1). Within the stromal cells, we identified fibroblasts, pericytes, and smooth muscle cells (SMC). Thymic fibroblasts broadly separated into capsular (Dpp4, Pi16) and medullary (Mmp9) fibroblasts[12]. The capsular fibroblasts included interlobular (CIFs) and two subsets of perilobular fibroblasts defined by expression of C7, Spry1 and Sparcl1 in perilobular 1 and in Aldh1a2 perilobular 2 subtype (Cap Peril 1 and 2). TECs separated into intertypical iTECs, mTECs, tuft-like mTECs and structural sTECs (Figs. 6, S5 and S6A, B and Supplementary Table 1).

The abundance of each cell type among mCherry+ labeled cells was semi-quantitatively compared to those in CRE-negative control

thymuses to determine enriched cell types (Figs. 6B and S6B). In agreement with the flow cytometry data, scRNAseq analyses revealed that in CD4-YY thymuses, mCherry+ cells were enriched for: B cells, DCs (cDC1, cDC2, pDC, aDC), ETPs, and fibroblasts (Cap Peril Fb and mFb in the scRNAseq data) (Figs. 6B and S6B). Additionally, scRNAseq data also detected αβ entry cells, iTEC, and mTEC within the interactome (Figs. 6B and S6B). Eosinophils were not detected as enriched in 8-week-old CD4-YY thymuses by scRNAseq (Fig. S6B).

TAM-treated Ubiq-YY thymuses harbor a range of senders (i.e., 13% DN, 44% DP, 21% SP4, 9% SP8, 6% ISPs) (Fig. S3Diii) different from that found in CD4-YY thymui.e., (i.e., 50% SP4s, 30% DP, and 7% SP8s) (Fig. 4C). Accordingly, the cellular composition of mCherry+ cells in Ubiq-YY thymuses showed clear differences with mCherry+ cells in CD4-YY thymuses (Figs. 6B and S6B). These interactome patterns further reinforced the specificity of the cellular interactomes that we identified.

CD4-YY thymuses harbored twice as many SP4 senders than DP senders (Fig. 4B, C) thus, most interactors should account for SP4 cellular interactions[3]. Most detected cellular interactors are known to anatomically locate preferentially to the medulla. This applies to known SP4 cellular interactors such as mTECs and mFb, which play critical roles in negative selection by expressing a plethora of peripheral tissue-restricted-antigens[3]. The role of iTECs is less explored as they were recently identified as a TEC subtype[84,85]. DCs play a crucial role in establishing central tolerance by presenting self-antigens derived from diverse sources[3,86,87]. This includes mTECs-derived self-antigens, which are essential for promoting tolerance to antigens expressed in a limited subset of peripheral tissues. These antigens are presented by activated DC1 (aDC1) and aDC2 subsets. In parallel, peripheral self-antigens carried into the thymic medulla by cDC2 cells enable tolerance to self-antigens that are not expressed by mTECs[17,88–91].

Specialized subsets of aDC2s (specifically CCR2+ and CX3CR1+ populations) transport and present serum-borne circulating antigens[92,93]. The role of pDCs in the establishment of central tolerance is less well characterized[94]. Likewise, B cells contribute to central tolerance through the presentation of self-antigens[3,70,95–98]. Upon entry into the thymus, B cells upregulate MHC-II and CD80 expression in a CD40 signaling–dependent manner. Unlike their peripheral counterparts, thymic B cells also express Aire[70,95–98]. Moreover, B cell receptor (BCR)-mediated antigen uptake and presentation has been also identified as an additional mechanism facilitating central tolerance[5,99]. In addition, thymic B cells can promote Treg induction[70,100].

A subset of identified cellular interactors known to localize preferentially near or within the thymic cortex[3,83] was also detected in our analysis. These included capsular perilobular fibroblasts, ETPs and iELpA (Fig. 6B). While the role of iELpA in thymocyte development remains unknown[23,83], capsular fibroblasts provide structural support within the thymus. The interplay between these fibroblasts and cTECs may contribute to T-cell development, although this relationship has yet to be fully elucidated[66].

Overall, our data revealed known and less characterized SP4 and DP cellular interactions in the thymus offering a new research avenue to investigate T cell biology. As previously indicated a small % of the identified interactions would correspond to SP8-interactors based on the presence of SP8 senders. Flow cytometry and scRNAseq methods complemented each other and exposed mostly overlapping cellular interactomes. scRNAseq uncovered a higher cellular heterogeneity offering a more in-depth picture on T cell interactomes.

## The global profile of cellular interactions is preserved with age in mice but the frequency of detected interactions decreases

Early in life the thymus undergoes a dramatic involution that reduces thymic cellularity in most vertebrates[84]. This is characterized by a reduction in the number of TECs, expansion of the perivascular space

and accumulation of adipose tissue that progressively impairs T cell development[101,102]. To unveil changes in the cellular interactome of SP4 cells with age, we isolated mCherry⁺ cells from the thymuses of 20-week-old *CD4-YY* mice. Thymuses from 20-week-old CRE-negative mice served as controls for assessing the enrichment in mCherry⁺ cells in *CD4-YY* thymuses.

The distribution of senders was comparable to thymuses in 8-week-old mice, although the % of SP4s was smaller (29 % of senders were SP4s; 43% were DP and 7% were SP8s) (Fig. S4Ai–iii). An average of 133,590 ± 63,514 sender cells were present in each thymus of *CD4-YY* mice at 20 weeks of age (Fig. S4Aiv).

mCherry⁺ recorded interactions were characterized by flow cytometry and scRNAseq (Figs. 7A and S6C, D). After quality control 23,505 single-cell transcriptomes were recovered and scRNAseq was able to capture similar populations as in 8 weeks. As expected, with a smaller number of captured single-cell transcriptomes compared to 8 weeks, some clusters were detected as combined populations. For instance, pDC and cDC2 are reported as a single pDC/cDC2 cluster and αβ entry and iELpA are reported together.

Surprisingly, the overall profile of the SP4/DP-interactomes was mostly preserved in 20-week-old mice compared to 8 weeks old[101–103] (Fig. 7B). Both flow cytometry and scRNAseq analyses showed that mCherry⁺ recorded interactions included B cells, DCs, ETPs, IL17⁺ γδ cells, innate lymphoid cells (ILC) and TECs (Fig. 7A, B). Interacting eosinophils were identified by flow cytometry in 8- and 20-week-old *CD4-YY* thymuses and by scRNAseq at 20 weeks, but not at 8-weeks old. Thus, we are not able to make conclusions on age-associated differences related to the eosinophil-SP4/DP interaction. Interestingly, with age we detected a spike on the frequency of mCherry⁺ interacting TECs (Fig. 7Bii), while the interactions with most stromal components decreased, as detected by both scRNAseq and flow cytometry (Fig. 7B). Likewise, interactions with other immune cells appeared less frequent (except for ETPs) and the overall identified interactions are also less frequent (Fig. 7B), as shown by a decreased % of mCherry⁺ labeled cells detected by flow cytometry, although it did not reach statistical significance ($p = 0.0544$) (Fig. 7Bi).

Importantly, these data support that the cellular interactions that maintain thymic homeostasis are being lost with age and add further evidence on how the thymic environment progressively degrades during thymic involution.

### The molecular crosstalk among SP4 and DP and their cellular interactome features pathways involved in survival, migration and differentiation

To investigate relevant molecular interactions among SP4 and DP and their cellular interactomes we employed the CellPhoneDB platform[22] (Fig. 8).

CellPhoneDB identified significantly enriched molecular pairs previously described to regulate migration, survival, and differentiation. These included CCL19-CCR7 and CCL21-CCR7 molecular pairs among TECs and SP4. mTECs and iTECs produced CCL19 and CCL21 chemokines known to facilitate SP4 migration to the medulla via CCR7[3] (Fig. 8). The CD40-CD40L pair respectively expressed by B-cells and SP4s was also enriched, where CD40L produced by T cells would stimulate Ag presentation in thymic B-cells[70,99,104]. Considering the prominent role of NOTCH pathway in T cell development[105], JAG1 produced by capsular perilobular fibroblasts 2 may influence DP differentiation via NOTCH1 as this molecular interaction was detected as significantly enriched among both cell types (Fig. 8). This influence would be larger in aged thymuses as the % of interacting capsular perilobular fibroblasts 2 increased in 20 weeks old thymuses (Fig. 7Bii). In 8-week-old mice, proliferating DP cells clustered together with proliferating DNs, while in 20-week-old thymuses a fraction of proliferating DP cells clustered separately (Figs. 6A and S6C). The expression patterns for the indicated molecular interactions of these populations together with those in quiescent DP cells are all shown in Fig. 8.

Several components of the SP4-interactome, including B cells, mFbs, γδ T17 and NKs, produced TGFβ, which is crucial for SP4 differentiation to Tregs[106,107] (Fig. 8). Pro-survival signaling as those triggered by CD80-CD28, CD86-CD28 and CD70-CD27 enriched molecular pairs would contribute to SP4 survival mediated by DC and B cells by increasing anti-apoptotic BCL-XL expression[108,109] (Fig. 8). Additionally, SP4-produced FLT3L could promote DC expansion via FLT3 Receptor (enriched FLT3L-FLT3R interaction among SP4 and many DC: pDC, cDC2 and aDC) to facilitate access to self-ligands. In this regard, FLT3L was shown to drive DC expansion in T cell-depleted hosts enhancing T cell accumulation by reducing the competition for self-ligands[110,111].

CellPhoneDB analyses also highlighted unexplored molecular interactions. SP4s expressed *Adgrl1*, a G-protein coupled receptor involved in cell adhesion and signal transduction[112], and mTECs produced multiple Teneurins (ADGRL1 ligands, including *Tenm1, Tenm3, Tenm4*), which are glycosylated-type II transmembrane proteins involved in neuronal development[112], with unknown role in SP4 biology (Fig. 8).

Interestingly, the patterns of expression related to these molecular interactions of SP4 and DP with their cellular interactome were mostly preserved in 20-week-old thymuses (Fig. 8B).

## Discussion

Most methods employed to date to identify cell interactions are either unable to directly detect actual cell-to-cell interactions (e.g., spatial transcriptomics), offer low-resolution and/or insufficient sequencing depth to capture rare cells or do not allow prospective isolation of interacting cells. Our unbiased genetic approach based on synNOTCH receptors fluorescently marked cells in actual physical contact with SP4 and DP cells and enabled their isolation and transcriptional characterization at single cell level to define their cellular interactomes.

synNOTCH receptors have been used in CAR-T cell therapies[36], to program self-organizing multicellular structures in vitro[113,114] and to detect in vitro cell interactions among breast cancer cell lines and brain stromal cells[115]. In vivo, Drosophila[38] and mouse[39] transgenic models have shown that synNOTCH can label cell contacts. However, until now, this technology has not been optimized and applied to taxonomize cellular interactomes in vivo. We have adapted its use to unveil the cellular interactomes that sustain thymic T cells by creating the Yin&Yang system and coupling it with flow cytometry and scRNAseq analyses.

Supporting the strength of our approach, we captured known critical cellular interactions, including those of SP4 with mTECs, mFb, DCs, and B cells[3,12,99,116], key in developing self-tolerance (Figs. 3–5). We also detected other less well-described SP4/DP cellular interactors such as iTECs[84,85] (Figs. 5–7). Importantly, we did not detect these cellular interactions in *Ubiq-YY* thymuses, which harbor a different profile of senders (Fig. S3D), reinforcing the specificity of these interactions.

Consistent with the role of cells like DCs and B cells as professional antigen-presenting cells (APCs)[117], treatment with an αMHC-II blocking antibody mostly erased the presence of mCherry⁺ cells from *CD4-YY* thymuses (Figs. 5B, C and S3F). Consistent with the interdependence of thymic cell populations in maintaining homeostasis[75,76], global MHC-II blockade broadly disrupted thymic interactions, underscoring our system's ability to detect both known and previously unrecognized cellular connections.

Likewise, doxycycline treatment, blocking tTA transcriptional activity, also abolished the presence of mCherry⁺ labeled cells, demonstrating the biological relevance and dynamic nature of the identified interactions (Figs. 5B, C and S3F).

Importantly, the observed reduction in the frequency of detected interactions during thymic involution in 20-week-old mice (Fig. 7B)

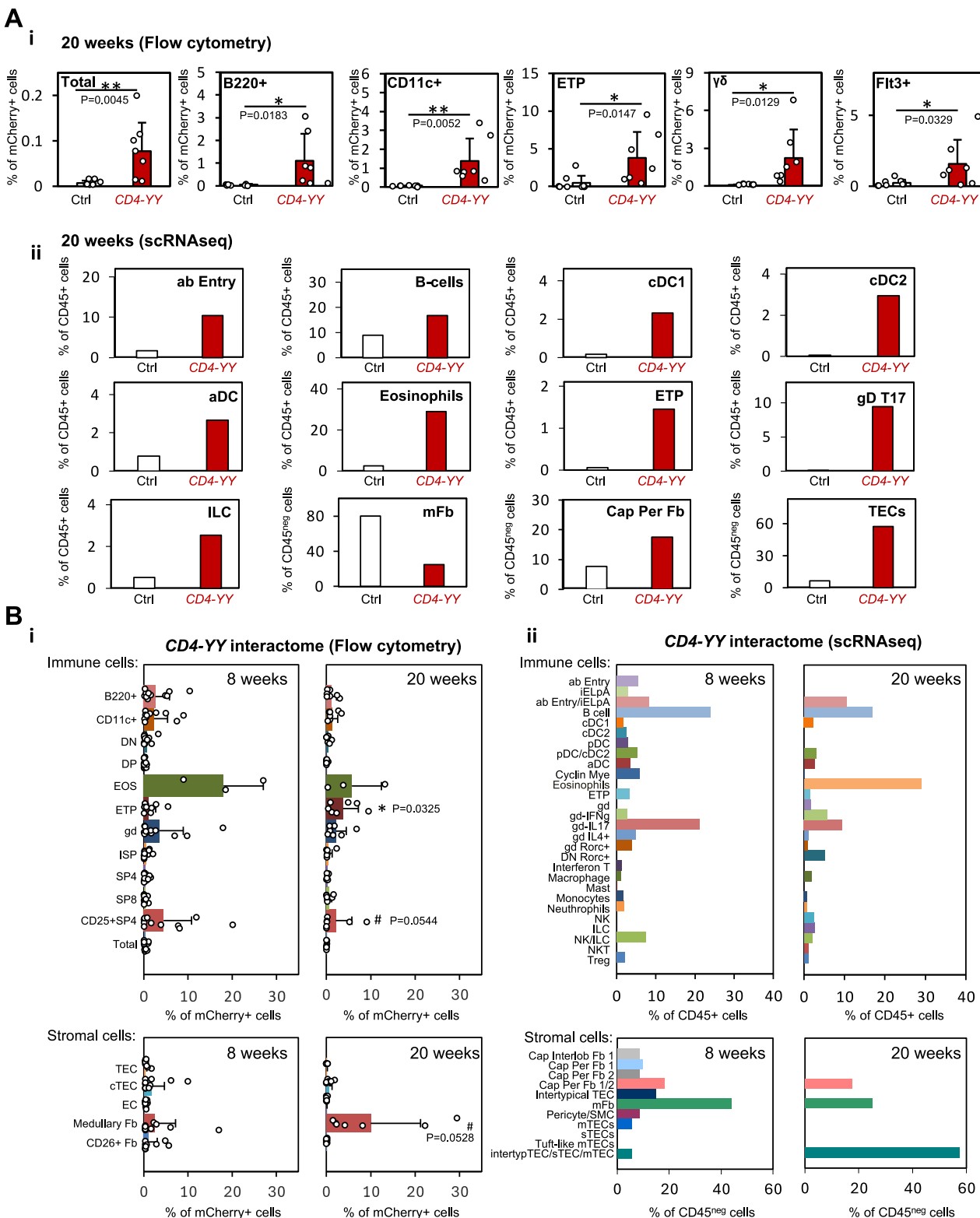

**A** **i** 20 weeks (Flow cytometry)

**ii** 20 weeks (scRNAseq)

**B** **i** *CD4-YY* interactome (Flow cytometry)

**ii** *CD4-YY* interactome (scRNAseq)

provides an additional layer of complexity to our understanding of how ageing progressively erodes the thymic cellular interactome that maintains thymic homeostasis. Whether this decline in cellular interactions represents a cause or a consequence of the gradual accumulation of adipocytes within the thymus, and how it contributes to the process of thymic involution, remains to be determined. This observation underscores the utility of our system as a powerful tool to investigate biologically significant processes, such as the mechanisms underlying age-associated thymic degeneration. Notably, the

identification of specific perturbed cellular interactions through this technology holds the potential to reveal therapeutic targets, thereby opening new avenues to prevent and/or reverse thymic involution.

Previous studies have shown that direct cell-to-cell interactions, mediated by the mechanical force applied by sender cells on interacting cells, are needed to activate synNOTCHs[36,37,40]. This process is similar to the activation of endogenous NOTCH receptors by NOTCH ligands[118,119]. Purified GFP is unable to trigger synNOTCH activation[36,37,39,40]. Thus, cell proximity without actual cell contact

**Fig. 7 | The cellular interactome that supports SP4 and DP cells in 20-week-old mice. A** Characterization of the cellular interactome (detected as mCherry⁺ cells) that supports SP4s and DPs in 20-week-old *CD4-YY* mice. **Ai** Flow cytometry characterization. *CD4-YY* mice (*n* = 7) and *YY^REC/REC TRE^Cherry/Cherry* CRE-negative control mice (Ctrl) (*n* = 9) were analyzed for mCherry⁺ enrichment (in three independent experiments). mCherry⁺ enrichment in *CD4-YY* compared to Ctrl thymuses underlines a cell-to-cell interaction. Each individual point represents an independent mouse. Rest of populations in Fig. S4B. **Aii** scRNAseq characterization. FACS-isolated mCherry⁺ CD45⁺ and mCherry⁺ CD45⁻ niche cells from *CD4-YY* thymuses (pooled from *n* = 11) and Ctrl mice (pooled from *n* = 18) were transcriptionally profiled at single cell in one experiment. The frequency of each cell cluster was calculated within mCherry⁺CD45⁺ and mCherry⁺CD45⁻ cells. Enriched clusters

indicate a cell interaction. Cell frequencies for other cell clusters in Fig. S6D. **B** Comparisons among the SP4/DP cellular interactome profiles detected in 8- versus 20-week-old *CD4-YY* thymuses. **Bi** Percentage of mCherry⁺ cells displayed by each cell population obtained via flow cytometry. *CD4-YY* data collated from Figs. 5C, Ai, S3F and S4Bii for age-related comparison. Thymuses from 6-8 weeks old *CD4-YY* mice (*n* = 12; from four independent experiments) and 20 weeks old *CD4-YY* mice (*n* = 7; three independent experiments) are shown. Each individual point represents an independent mouse. **Bii** Percentage of CD45⁺ and CD45⁻ cells within sorted mCherry⁺ identified by scRNAseq. Data collated from Figs. 6B, Aii, S6B and S6D for comparison. Means and standard deviations are depicted. **Ai, Bi:** Unpaired two-tailed t-test. ****p < 0.0001, ***p < 0.001, **p < 0.01, *p < 0.05, #p < 0.1.

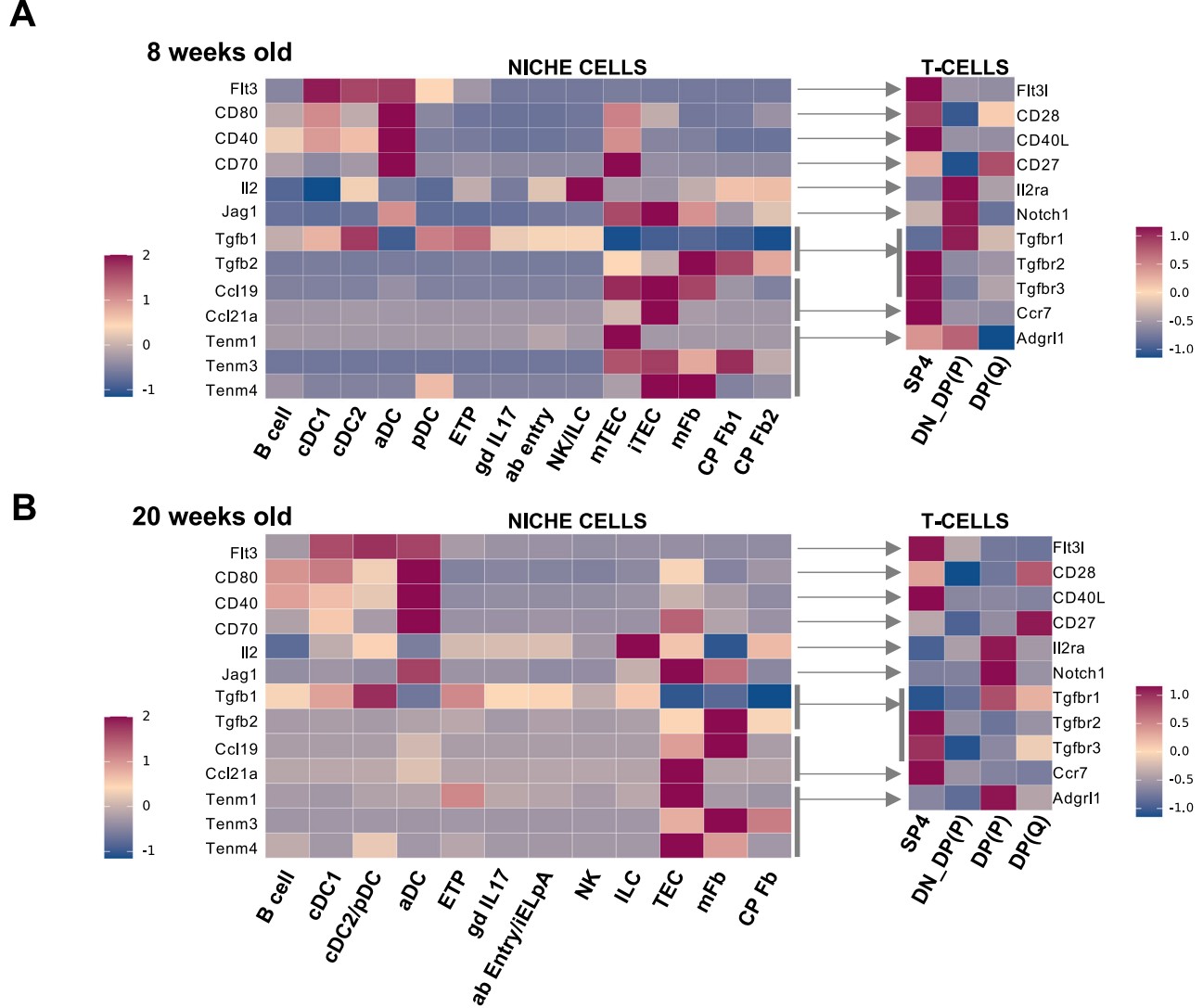

**Fig. 8 | Molecular interactomes for SP4 and DP cells.** Heatmaps summarizing inferred molecular interactomes for SP4 and DP cells in 8 weeks (**A**) and 20 weeks-old thymuses (**B**) based on CellPhoneDB analyses. Cell arrows indicate molecular

pairs (ligands – receptors). (P): proliferating. (Q): quiescent. DN_DP (P): cluster of proliferating DN and DP cells.

cannot drive synNOTCH activation. This likely impacts the type of interactions that the Yin&Yang system can record and our results do not exclude the presence of other cellular interactions. Importantly, our system is capable of detecting brief cellular interactions (<1 h) (Fig. S3B) such as those involved in the induction of central tolerance[17] and positive selection[47]. Notably, the H2B-mCherry-based interaction-recording system exhibits high stability (persisting for at least 6 days and five cell divisions, Fig. 2C), which undoubtedly facilitates the

detection of such transient events. Indeed, even three intermittent interactions of 30 min each were sufficient to generate detectable mCherry signals (Fig. S3B).

On the potential concern that mGFP-synNOTCH binding may preclude cell dissociation, we have observed that in *Ubiq* thymuses ~70% of labeled receiver cells are nearby but not in physical contact (Figs. 3Biii and S3E), demonstrating that prior ligand-receptor binding does not block the dissociation of interacting cells.

Other approaches to record direct cell-to-cell contacts have been developed in neurobiology, including rabies viruses, optogenetics, and split forms of fluorescent proteins (GFP and CFP)[35] to study synaptic interactions. In the thymus, enzymatic transfer of a labeled substrate between ligand and receptor (LIPSTIC) was recently reported to detect cell interactions[120,121]. Rapid enzymatic transfer of substrates among cells allows LIPSTIC to detect brief cell contacts, as those during Ag presentation. Substrate-acceptor cells can be detected within a pool of scRNA-sequenced cells via DNA-barcoded antibodies. However, LIPSTIC does not allow pre-sorting of interacting cells decreasing its sensitivity for rare cell populations. Additionally, LIPS-TIC only allows tracking cell-interactions for short periods of time (substrate labeling is detectable only for 4–6 h post-administration)[121]. By contrast, H2B-mCherry-based recording of cell interactions in our YY mice is stable (6 days and 5 cell divisions), allowing for long-term tracking of cellular interactions (Fig. 2C). Moreover, our system enables detection of repetitive brief interactions shorter than one hour (Fig. S3B). Both mouse models could potentially be used simultaneously as they both depend on CRE expression to induce senders (in YY mice) and substrate donors (in LIPSTIC mice), complementing each other.

Overall, our Yin&Yang genetic-based method for the identification of cellular interactomes in vivo provided an unbiased taxonomy of cell-to-cell interactions that support T cells in the thymus. Our study paves the way to unveil molecular and cellular interactions in T-cell development and in thymus involution.

### Limitations of the study

The lack of a CRE line that specifically targets SP4, DP, and SP8 cells is a limitation. Such a model would allow to more precisely define their specific interactomes. The moderate recombination efficiency of the $YY^{REC}$ allele can complicate the applicability of the YY system to identify the cellular interactomes of other less frequent cell types in the thymus. Accordingly, the efficiency of recombination driven by any other CRE line in producing sender cells will need to be evaluated and optimized. For instance, longer tamoxifen exposures are expected to increase recombination efficiency in CRE-ERT2 lines[122]. If applying the YY system to other tissues different from the thymus, the background expression from the $TRE^{Cherry}$ reporter allele will need to be assessed.

## Methods

### Study design

The aim of this study was to determine the cellular interactomes that support thymic SP4, DP, and SP8 cells and how they change with age. We generated and optimized the Yin&Yang mouse model to record actual cell-to-cell interactions in vivo in the thymus. We then combined the Yin&Yang model with the *CD4-Cre* allele which mostly rendered thymic DP and SP4 sender cells. A small fraction of SP8 sender cells was also induced. Sender cells induced fluorescent labeling of interacting cells in an unbiased manner by activation of an orthogonal synthetic pathway based on synNOTCH activation. Interacting cells were identified by flow cytometry and scRNAseq. We validated that these interactions are biologically relevant and dynamic by treatment with αMHC-II blocking antibody and doxycycline (which blocks the orthogonal pathway). Analysis of older mice showed that even though the cellular interactome profiles are maintained with age, the frequency of the interactions is reduced. CellPhoneDB served us to identify molecular interactions among DP and SP4 and their cellular interactomes.

### Animals

All experiments involving mice were performed under Queen Mary University of London Veterinary oversight with UK Home Office authorization (Project License number: PP7312641, Project Licence holder: Miguel Ganuza). and comply with all relevant ethical regulations regarding animal research. Ubiquitin$^{+/ERT2-Cre}$ [B6.Cg-Ndor1Tg(UBC-cre/ERT2)1Ejb/1J, stock #007001][48], *R26-Flipase* [B6 ROSA26Flpo; B6.129S4-*Gt(ROSA)26Sor^{tm2(FLP*)Sor}*/J, stock #012930][123]; *E2a-Cre* [B6.FVB-Tg(EIIa-cre) C5379Lmgd/J; stock #003724][42], *Col1a1-TRE-H2BmCherry* [*Col1a1-tetO-H2B-mCherry*; stock #014602][44], *CD4-Cre* [B6.Cg-Tg(Cd4-cre)1Cwi/BfluJ; stock #022071][41] mice were used (Jackson Laboratory (Bar Harbor, Maine) and housed in a specific pathogen-free facility/SPF. The generation of the Yin&Yang mice is described in Fig. 1A. The *YY* transgene was chemically synthesized and cloned into intron 1 of *ROSA26* locus in reverse orientation in C57BL/6N mice by Taconic-Cyagen. To engineer the targeting vector, homology arms were generated using BAC clone as a template and Cas9 and gRNA were co-injected along with the targeting vector into fertilized eggs to produce knock-in mice. The pups were genotyped by PCR and analyzed by sequencing. Animals were maintained under specific pathogen-free conditions. All mice were on the C57BL/6 genetic background. Both male and female mice at the age of 6–25 weeks were employed in this study. Mice were housed in a temperature-controlled room ($22 \pm 2\,°C$) with a relative humidity of ($50 \pm 10\%$). A 12-h light/dark cycle was maintained (lights on at 07:00). Mice were provided with *ad libitum* access to food and water, and nesting material was provided for environmental enrichment. Animals were euthanised by Schedule 1 Standard Methods of euthanasia [humane methods for killing animals under the UK's Animals (Scientific Procedures) Act 1986 (ASPA)], including cervical dislocation and $CO_2$ inhalation.

### Cell lines

Mouse embryonic fibroblasts (MEFs) were derived from E13.5 embryos in our own laboratory. We generated *E2a-Cre*$^{+/Cre}$ *YY*$^{+/mGFP}$ Yin sender MEFs and *Col1a1-TRE-H2BmCherry*$^{Cherry/Cherry}$ *YY*$^{REC/REC}$ Yang receiver MEFs. Genotypes were obtained post-isolation by Polymerase chain reactions (PCRs) as detailed in the genotyping section at the time of isolation. The sex of the embryos was unknown.

### Genotyping

PCRs were performed employing HotSar Taq DNA Polymerase (Qiagen) following the manufacturer's instructions. PCR conditions: ($95\,°C$, 2′); (($94\,°C$, 30″; $56\,°C$, 30″; $72\,°C$, 30″] × 35); ($72\,°C$, 10′). Primers: Cre F1 (5′-CTGTTACGTATAGCCGAAAT-3′), Cre R1 (5′-CTACA CCAGA-GACGGAAATC T-3′)[124,125]. CRE-positive PCR band: 203 bp. To detect the *ROSA26*$^{FLPO}$ allele, we employed the following primers obtained from Jackson Laboratories: oIMR8052 (mutant reverse) (5′- GCG AAG AGT TTG TCC TCA ACC -3′), oIMR8545 (mutant reverse) (5′- AAA GTC GCT CTG AGT TGT TAT -3′), oIMR8546 (mutant reverse) (5′- GGA GCG GGA GAA ATG GAT ATG -3′), WT band (650 bp), Mutant band (340 bp). To detect *R26-YY*$^{iYY}$ allele we used F4 and R4 primers. F4 (5′-CGTCTCCGTCCAGCTCAACCAGG -3′), R4 (5′- CGCATTGTCTGAG-TAGGTGTCATTC -3′), *R26-YY*$^{iYY}$ PCR band (F4 + R4): 285 bp.

To detect *R26-wt* allele, we employed F7 and R7 primers. F7 (5′-AAGCACGTTTCCGACTTGAGTTG -3′), R7(5′-GGGTGAGCATGTCTT-TAATCTACC -3′). Wildtype allele PCR band: 607 bp. To detect *R26-YY*$^{REC}$ we used primers F8 + R8, F8(5′- CTACGAGTTGCACATCAGCCATC -3′),R8(5′- GGCAACGTGCTGGTTATTGTG -3′). *R26-YY*$^{REC}$ allele is detected as a 259 bp PCR product. The use of F8 + R8 is also able to detect *R26-YY*$^{iYY}$ and renders a 1136 bp band. To detect *R26-YY*$^{mGFP}$, we employed primers F4 and R8. *R26-YY*$^{GFP}$ is detected as a 274 bp PCR band. Please see schematic in Supplementary Fig. 1A depicting the location of primers relative to the Yin&Yang allele.

### PB analysis

Mouse peripheral blood was collected in K2 EDTA-coated capillary tubes (microvette CB300, Sarstedt) from the lateral vein of the tail, lysed in red blood cell lysis buffer (Zen-Bio), and analyzed for the expression of GFP. To assess cellular viability, 0.1 μg/mL 4′,6-diamidino-2-phenylindole (DAPI) staining was used.

## Tamoxifen, doxycycline and blocking antibody delivery in vivo

To activate Ubiquitin-CRE-ERT2, 6–7 weeks old mice were treated for 11 consecutive days by oral gavage (20 ga × 38 mm plastic feeding tubes, Instech Laboratories) with 2 mg of tamoxifen (TAM, T5648, Sigma-Aldrich) suspended in 90% sunflower seed oil (Sigma-Aldrich) and 10% ethanol (Fisher BioReagents). Mice were always treated with 200 μL of 10 mg/ml TAM. Thymuses were analyzed 15 days post-TAM treatment initiation.

To block the transcriptional activity of tTA, doxycycline was administered in the drinking water at a concentration of 1 mg/mL for 3 weeks.

Blocking antibodies for MHC-II (clone M5/114, InVivoMab, cat#BE0108) were administered intraperitoneally. MHC-II was provided at a daily dose of 150 μg per mouse for 14 days.

## Flow cytometry analysis

Flow cytometry analyses were performed on a FACSymphony or a LSR Fortessa I flow cytometer (BD Biosciences, San Diego, CA). Data were collected with BD FACSDiva Software (version 8.0.1) (BD Biosciences, San Diego, CA) and analyzed with FlowJo version 10.8.0 (BD Life Sciences).

## Thymus analysis

Thymuses were collected from carbon dioxide euthanised animals and dissociated with a working solution of 1 mg/mL collagenase/dispase (ROCHE) containing 0.1 U/mL of collagenase and 0.8 U/mL of dispase for 30 min at 37 °C. After lysis with red blood cell lysis buffer (Zen-Bio), cells were stained and analyzed to identify early thymic progenitors [ETPs, Lin$^-$(CD11b$^-$CD19$^-$NK1.1$^-$Ter119$^-$)cKit$^+$CD44$^+$CD25$^-$CD24$^{-/+}$][59–61], CD45$^+$B220$^+$ (containing B cells), CD45$^+$Flt3$^+$CD4$^-$CD8$^-$CD25$^-$cKit$^{-/intermediate}$ (containing DC precursors)[58], thymocytes[126] [including single-positive CD4 SP4 cells: CD19$^-$CD11b$^-$NK1.1$^-$TER119$^-$CD8$^-$CD4$^+$; CD25$^+$SP4: CD19$^-$CD11b$^-$NK1.1$^-$TER119$^-$CD8$^-$CD4$^+$CD25$^{high}$ enriched for T regulatory cells[127]; DP cells: CD19$^-$CD11b$^-$NK1.1$^-$TER119$^-$CD8$^+$CD4$^+$; pre-positive selection DPs: CD69$^-$TCRαβ$^{LOW}$[31,50–55]; early post-positive selection DPs (CD69$^+$TCRαβ$^{LOW}$)[31,50–55]; co-receptor reversing DPs (CD69$^+$TCRαβ$^{HIGH}$)[31,50–55]; single-positive CD8 SP8 cells: CD19$^-$CD11b$^-$NK1.1$^-$TER119$^-$CD8$^+$CD4$^-$; immature single positive CD8 ISP cells: CD19$^-$CD11b$^-$NK1.1$^-$TER119$^-$CD8$^+$CD4$^-$CD24$^+$TCRβ$^-$; CD4$^-$CD8$^-$ DN cells: CD19$^-$CD11b$^-$NK1.1$^-$TER119$^-$CD8$^-$CD4$^-$; and γδ T cells: CD19$^-$CD11b$^-$NK1.1$^-$TER119$^-$ TCRβ$^-$TCRγδ$^+$CD8$^-$CD4$^-$[128]] and thymic stromal cells[3] [including CD45$^+$CD11c$^+$ cells: containing ~75% of DCs and ~25% of macrophages[56,57]; medullary fibroblasts: CD45$^-$CD31$^-$EPCAM$^-$Gp38$^+$CD26$^-$[63,65,66]; capsular fibroblasts: CD45$^-$CD31$^-$EPCAM$^-$CD26$^+$[63,65,66]; endothelial cells[129]: CD45$^-$EPCAM$^-$CD31$^+$; CD31$^+$EPCAM$^+$ cells: CD45$^-$CD31$^+$EPCAM$^+$; thymic epithelial cells TEC: CD45$^-$CD31$^-$EPCAM$^+$[130] and cortical TEC: CD45$^-$CD31$^-$EPCAM$^+$UEA-1$^-$Ly51$^+$[62–64,131]]. The following antibodies were employed for the identification of thymic progenitors and B cells: c-Kit-APC Fire750 (2B8; cat. #105838), CD34 PE (SA376A4; cat #152204) (1:50), CD135-Biotin (A2F10; cat #135308) (1:50), streptavidin-BV605 (cat #405229), CD135-APC (A2F10; cat #135309) (1:50), CD45.2-Brilliant Violet 510[104], CD4-PerCP (GK1.5), CD8-PerCP (53-6.7), CD127-PECy7 (A7R34), CD25-Brilliant Violet 605 (PC61; cat #109838), Sca-1-BV711 (D7; cat #108131) and B220-Alexa Fluor 700 (RA3-6B2; cat #103232. For the characterization of thymocytes the following antibodies were used: CD19-PerCP (6D5; cat #115532), CD11b-PerCP (M1/70; cat #101230), NK1.1-PerCP (PK136; cat #108726), TER 119-PerCP (TER-119; cat #116226), CD4-Alexa Fluor 700 (RM4-5; cat #100536), CD8-PECy7 (53-6.7; cat#100721), CD25-Brilliand Violet 605 (PC61; cat #102036), CD44-APCCy7 (IM7; cat #103028), CD24-PE (cat #101808), TCRbeta-Brilliant Violet 510 (H57-597; cat #109234) or TCRβ-APC (H57-597; cat# 109212), TCRγδ-Brilliant Violet 650 (GL3; cat #100651), cKit-Brilliant Violet 785 (2B8), CD127-Brilliant Violet 711 (A7R34), D45RB-APC (C363-16A), CD5-Brilliant Violet 605 (53-7.3; cat #105841), CCR4-PE (2G12; cat #131204), CD69-APCCy7 (H1.2F3; cat #104526), CD62L-Brilliant Violet 650 (MEL-14; cat #104453) and CCR7-Brilliant Violet 785 (4B12; cat #120127). To identify thymic stromal cells, the following antibodies were utilized: CD45.2-Brilliant Violet 510 (104; cat #109838), EpCam-PerCPCy5.5 (G8.8; cat #118220), CD31-Brilliant Violet 711 (390; cat #102449), CD34-PE (SA376A4; cat #152204) (1:50), Gp38-APCCy7 (8.1.1; cat #127418), CD26-APC (H194-112; cat #105841), CD11c-A700 (N418; cat #117319) and streptavidin-Brilliant Violet 605 (cat #405229). For the identification of eosinophils, the following antibodies were employed: CD45-Brilliant Violet 605 (30-F11; cat #103139), Siglec-F (CD170)-PE (S17007L; cat # 155506), and CD11b-Brilliant Violet 785 (M1/70; cat #101243). All antibodies (Biolegend) were employed at 1:200 unless specified and 0.1 μg/mL DAPI was used to exclude dead cells.

## Immunofluorescence and confocal microscopy

MEFs were cultured on poly-L-lysine (0.1 mg/mL; #P2636, Sigma-Aldrich) coated glass coverslips and fixed for 20 min in 4% paraformaldehyde (PFA; #043368.9 L, Thermo Fisher Scientific). Cells were permeabilized for 30 min with 0.2% (w/v) Triton X-100 (TX-100; #T8787, Sigma-Aldrich) and incubated 40 min with Alexa Fluor 647 Phalloidin (1:200; #A7906, Invitrogen) in 1% BSA/PBS, and 10 min with DAPI (1:1000). Samples were mounted in slides employing Fluoromount-G (#00-4958-02, Invitrogen).

Seven-to-eight-week-old mice were intracardially perfused with 30–50 mL of DPBS 1X (Dulbecco's, pH 7.4; #14190144, Gibco), followed by 50 mL of 4% PFA. Thymuses were isolated, post-fixed in the same fixative at 4 °C overnight, cryoprotected for 2 days in 30% sucrose (#S0389, Sigma-Aldrich), embedded in optimal cutting temperature media (OCT; #23-730-571, Fisher Scientific) and frozen for sectioning. Tissue slices were permeabilized and blocked for 1 h (2% TX-100; 2% normal goat serum; #AB7481, Abcam) under shaking at RT. Slides were then incubated with primary antibodies anti-c-Myc-tag (1:500; cat #Vli47, MaineHealth Institute for Research) or anti-GFP (1:100; #AB183734, Abcam) overnight at 4 °C on a wet chamber, and secondary antibodies Alexa Fluor 555 or 488 (1:200; cat #AB150078 and #AB150077, respectively, Abcam) at RT for 2 h. A final 10 min incubation with DAPI (1:1000) was performed. MEFs and thymuses were imaged on a Zeiss LSM880 Confocal Microscope (Zeiss, Oberkochen, Germany) using either a ×40/1.3 or ×60/1.4 Oil DIC Plan-Apochromat objective. Images were processed with FIJI-ImageJ (1.54 f).

## Histopathology and immunostaining

For routine histological analysis, tissues were fixed in 10%-buffered formalin (Sigma) and embedded in paraffin. Haematoxylin and eosin (bluing reagent) (#760-2021 and #760-2037, respectively; Roche Diagnosis, Ltd) staining was performed automatedly on a Leica Autostainer XL and mounted in a Leica CV5030 Cover-slipper (Leica, Wetzlar, Germany).

Immunohistochemical processing and staining were performed either in a Ventana Discovery XT or a Ventana Discovery Ultra machine (Roche Diagnosis, Ltd). Deparaffination, antigen retrieval, blocking and detection were done as recommended by the manufacturer using the Discovery ChromoMap DAB Kit (#05266645001, Roche). Primary antibodies anti-c-Myc-tag (1:1000; cat #Vli47, MaineHealth Institute for Research) or anti-GFP (1:1600; cat #2555S, Cell Signalling) were incubated for 60 min, followed by Omnimap anti-Rabbit HRP during 16 min, and revealed with DAB/$H_2O_2$ and copper for 8 min and 5 min, respectively. Slides were then counterstained with haematoxylin and bluing reagent for 8 min each and mounted with permanent media. Images were acquired with the slide scanner NanoZoomer S210 (Hamamatsu) using a dry 40× objective and in a semi-manual mode, stablishing at least 5 focus point locations on each tissue. Images were reviewed and exported with the NDP.view2Plus software provided by Hamamatsu.

## Single-cell transcriptomics- sample preparation

For single-cell RNAseq analysis, thymuses from ≥8 mice per strain were collected and processed as indicated before. The resulting cell suspension was stained with CD45.2-BV510[104] (Biolegend). As CD45⁻ stromal cells in the thymus constitute a small cellular fraction, we isolated mCherry⁺ CD45⁻ and mCherry⁺ CD45⁺ cells by FACS on a BD FACSAria II (BDBiosciences, San Diego, CA) and mixed them at ~1:1 ratio to enable the identification of stromal cell clusters.

About 14,000 cells from each group (mCherry⁺CD45⁺ and mCherry⁺CD45⁻) were loaded together with ~1000 GFP⁺ sender cells into the 10× Genomics Chromium instrument for GEM (gel bead in emulsion) generation, followed by cDNA preparation and gene expression library construction (Chromium Next GEM Single Cell 3′ Kit v3.1, 16rxns PN-1000268). ~10,000 cells were captured per sample and sequenced to >18,000 reads/cell to obtain gene expression profiles.

## Quantification and statistical analysis

**Statistics and reproducibility.** Summary statistics are reported as means ± standard deviation from at least three different values unless otherwise indicated. No randomization method was used. A Shapiro-Wilk normality test was first performed. Then, statistical significance between two different groups was determined using a two-tailored Student's t-test or Mann-Whitney's U test, and correlation between groups by Pearson's or Spearman tests. Tukey´s multiple comparison test was performed at a level of 0.05 to test for differences among the means of three or more groups. The exact *P*-value was provided for each test. *P*-values < 0.05 were considered statistically significant. Sample size and number of experiment replicates are detailed in each Figure Legend. Analyses were conducted in GraphPad Prism 9. No statistical method was used to predetermine sample size. Data exclusion was only applied in the case of the thymus analysis by flow cytometry, presented in Figs. 4–5, 7Ai, S3F and S4Aii, where animals non-carrying sender cells were removed from the analysis.

Animals were allocated into experimental groups according to genotype. Although no specific methods were used for blinding, blood and thymus samples were collected from mice by one individual and then analyzed by flow cytometry and scRNAseq by different individuals. Experiments were reliably reproduced. In vivo results on the specific expression of the YY system were verified by IHC, confocal microscopy, and IF. In vitro results on the ability of the YY to record cell interactions were verified employing H2B-mCherry as a fluorescent-based recording method and validated in vivo in *Ubiq-YY* mice. Major findings on the interactomes of SP4 and DP cells employing the Yin&Yang genetic system identified by flow cytometry were supported by scRNAseq.

## Single cell transcriptomics—Determining abundance of clusters within mCherry⁺ cells.

mCherry⁺ CD45⁻ and mCherry⁺ CD45⁺ cells were mixed at ~1:1 ratio. As CD45⁻ mCherry+ cells were manually enriched in the samples before sequencing to enable capturing of rare CD45⁻ stromal cells, the frequencies of CD45⁻ clusters were evaluated within mCherry⁺CD45⁻ cells to determine the abundance of the identified clusters within each strain. Similarly, the frequencies of CD45⁺ clusters were assessed within mCherry⁺CD45⁺ cells. Frequencies for SP4, DP, and DN clusters are not shown as GFP⁺ senders were manually enriched before sequencing affecting their abundance, and were excluded from the analysis.

Cell clusters were annotated for mCherry⁺ sorted cells for each experimental mouse strain (Fig. 6). The cellular frequencies of identified clusters in *CD4-YY* were compared with thymuses isolated from *YY^REC/REC^ TRE^Cherry/Cherry^* control mice to identify enriched interacting populations (Figs. 6B, 7Aii, S6B and S6D) and to those in *Ubiq-YY* thymuses (Figs. 6B and S6B).

**Analysis of single-cell transcriptomics data.** Single-cell RNA-seq data were pre-processed using the 10× Genomics Cell Ranger (v2.0) pipeline. Downstream analyses were performed with Seurat v4[132]. The raw gene expression matrices were filtered using the following quality control criteria[1]: >300 genes[2]; <10% mitochondrial reads. Ribosomal and mitochondrial genes were discarded. The datasets were concatenated into a single gene expression matrix. The data were normalized with a scale factor of 10,000 and log1p-transformed. We extracted 2000 highly variable genes (HVGs). The percentage of mitochondria and number of detected genes were regressed out. A neighborhood graph (KNN) was built using the resulting 30 principal components to perform leiden clustering and UMAP visualization.

To define major cell types, cells were clustered using the Leiden method (resolution parameter $r = 0.8$). Differentially expressed genes were identified for each cluster using the Wilcoxon rank-sum test with Benjamini−Hochberg *p*-value correction in Seurat. The transcriptomes were partitioned into 8 major cell types (thymic epithelial−TEC, stromal, myeloid, mast, B cells, natural killer (NK)/innate lymphoid cells (ILC), double negative double positive, single positive CD8 and CD4 T cells, Tregs, and γδ T cells by comparing differentially expressed genes and canonical markers from the literature.

Subsequently, the same integration and clustering analysis was applied iteratively to the cells of each major cell type separately to identify and annotate cell states. For the T cell analysis, genes associated with the dissociation effect were removed from the dataset[133]. Scrublet (v0.2.3)[134] was run per sample to identify potential doublets. Clusters with high numbers of doublet cells were removed by checking for potential doublets identified by Scrublet and using the expression of markers of more than one cell type.

Cell-cell communication networks were inferred using CellPhoneDB (v3.1.0, database v4.0.0)[22]. Ligand-receptor interactions satisfying the following criteria were selected[1]: all ligands and receptors were expressed in at least 10% of the cells of each cell state[2]; the ligand-receptor interactions between two cell states were inferred using the statistical analysis method in CellPhoneDB with a *p*-value threshold of 0.05[3]; ligand-receptor interactions were pruned based on mean expression levels.

## Reporting summary

Further information on research design is available in the Nature Portfolio Reporting Summary linked to this article.

## Data availability

All unique reagents generated in this study (Yin&Yang mouse strains and immortalized cell lines) are available with a completed Materials Transfer Agreement. Yin&Yang mice will be deposited in mouse repositories to enable easier access following publication. scRNAseq data generated in this study have been deposited in the NCBI-SRA database under BioProject ID: PRJNA1276306. [https://www.ncbi.nlm.nih.gov/bioproject/?term=PRJNA1276306]. Additional underlying data supporting the findings of this study, including raw flow cytometry (FCS) files, are available from the corresponding authors upon reasonable request. Requests for access should be submitted by email to the corresponding authors and include a brief description of the intended use. Access will be granted for academic, non-commercial research purposes, subject to appropriate data use and not redistributed. Source data are provided with this paper.

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

## Acknowledgements

We thank Ellen Rothenberg (California Institute of Technology) and Shannon McKinney-Freeman (St. Jude Children's Research Hospital) for critical discussions on the manuscript; Nasrine Metic (QMUL) for advice on computational analyses; Manuela Terranova Barberio, Stephen Rogers, Foram Vaidya and Athar Diwani-Massarwa for FACS support; Jordan Chattenton, Harmony Blythin, Samuel Potter, Michelle Murphy, Jordi Tremoleda and BSU department at QMUL for their assistance with animal work; Eva Wozniak, Paul Stevens and Charles Mein at the Genome Centre – Blizard Institute (QMUL) for DNA library preparation; and Findlay Bewicke-Copley for assistance with SRA files. Funding: American Society of Hematology. Global Research Award, ASH GRA 2021 (M.G.); Barts Charity. Research Project Grant G-002877 (M.G.); Barts Charity—The Rising Stars Programme, MGU0459 (M-.G. and M.E.); Greg Wolf Fund (M.G.); Kay Kendall Leukemia Fund. KKL1444 (M.G.); Leukemia UK. John Goldman Fellowship, 2020/JGF/001 (M.G.); Medical Research Council. MRC Career Development Award, MR/V009222/1 (M.G.); Cancer Research UK. PhD studentship (F.K.); Leukemia UK. 2022/JGF/001 (M.E.); Cancer Research UK. Establishment Award, RCCCEA/100003 (M.E.); Medical Research Council. PhD studentship (E.S.).

## Author contributions

Conceptualization: M.G., Methodology: R.S.L., M.G., and M.E., Investigation: R.S.L., A.J.P., N.S., J.B., F.K., and E.S., Formal analysis: R.S.L., M.G., M.E., and A.J.P., Funding acquisition: M.G., Project administration: M.G., Supervision: M.G., M.E., Writing: M.G., R.S.L., M.E., and D.P.

## Competing interests

The authors declare no competing interests.
