## [Transparent Peer Review file · Nature Communications]

Unbiased recording and identification of thymic cellular interactomes using synthetic Notch receptors.

Corresponding Author: Dr Miguel Ganuza

Version 0:

Reviewer comments:

Reviewer #1

(Remarks to the Author)

In this manuscript, Sánchez-Lanzas et al. describe the generation of a Yin-yang (YY) mouse system with which to detect cellular interactions between thymocytes and other cells in the thymus microenvironment in an unbiased manner. Detecting the cellular interactome and molecular mediators of thymocyte-stromal interactions is highly significant because these interactions govern differentiation and selection of CD4+ and CD8+ T cells with a diverse but self-tolerant TCR repertoire. In the YY system, GFP is expressed by “sender” cells due to Cre-mediated excision of a STOP codon preceding a membrane-bound GFP. Non-sender cells ubiquitously express a synthetic notch receptor comprised of a single-chain antibody specific for GFP linked to the NOTCH transmembrane and partial intracellular domain followed by a tetracycline transactivator (tTA). All cells in the YY mouse carry an allele expressing mCherry behind a tetracycline responsive element. Thus, when the “receiver” cells bind to GFP on sender cells, the synthetic Notch receptor is cleaved, releasing to tTA to induce mCherry expression. The authors thoroughly characterize this elegant genetic system, demonstrating that mGFP expression is restricted to Cre-expressing cells, that the receptor for receiver expression is widespread across cell types and tissues, and that mCherry expression (at least high expression levels) occurs only when mGFP+ donor cells are present. Importantly, they also show that interactions of at least 2 hours are needed to detect receiver cells and that mCherry expression persists across multiple cell divisions for at least 9 hours after induction. The authors then use CD4-cre to drive mGFP expression mainly in a mixture of SP4 (~50%) and DP (~30%) thymocytes, and they identify the receiver mCherry+ cells using both flow cytometry and scRNA-seq. With both modalities, they identify interactions of sender cells with known thymic APCs, such as B cells and DCs, as well as unexpected cell types, including ETPs, gamma delta T cells, fibroblasts, and ECs. Comparisons of interactomes between 5 week-old and 20 week-old mice reveal that the distribution of receiver cells is largely maintained at this stage of thymus involution, but that interactions are less frequent. Finally, they use CellChat DB analysis of their scRNAseq data to identify putative molecular mediators of cellular interactions, finding known signals, like CD40-CD40L, as well as novel candidate molecular interactions, such as Tenm1,3,4 with Adgrl1. Strengths of the paper include the elegant system and experiments carefully evaluating the specificity and kinetics of detecting receiver cells. However, it remains unclear based on the controls presented that the interactions detected are physiologically relevant for T-cell development and/or selection. Furthermore, there are concerns about overinterpretation of results, particularly with respect to assumptions about whether SP4 or DP mediate the detected interactions and whether these interactions occur in the cortex or medulla. Detailed major and minor concerns follow:

Major

1. Because identification of sender and receiver cells relies on accurate flow cytometry gating, it is critical that the cell populations in the thymus are more accurately defined by flow cytometry. Figure S2Ai shows a gating strategy to identify LMPP, CLP, and ETP in the thymus. However, LMPP and CLP are present in bone marrow, not thymus, while ETP are present in the thymus. So it is not clear what populations or organs are being analyzed, as the legend indicates that the cells shown are from the thymus. It's possible that the Flt3+ cells are DC precursors in the thymus (doi 10.1002/eji.201141728). ETP are typically defined as lineage- cKIT+ CD44+CD25-, as later acknowledged by the authors, with the most immature thymocyte progenitors expressing Flt3. The LMPP gate and CLP gate do not look clean based on CD127 staining (if this is a BM analysis, which I'm not sure it is). Please gate the populations more precisely in the thymus, which will be critical for proper analysis of sender and receiver cells. In S2Aii, Treg should be defined as CD4SP thymocytes (CD3+) that express CD25 and Foxp3. The gate on CD25 does not clearly delineate the Treg population. In S2 A iii, please correct the x-axis in the 2nd flow cytometry panel (CD11c); however, DCs in the thymus should be gated

as CD11c+MHCII+ cells that do not express F4/80 to avoid macrophages. Also, Gp38/podoplanin is expressed by both mesenchymal cells and TECs in the thymus, so the CD45-Gp38+ cells will be a mix of these cell types. If a CD26 gate is used to define the “CD26+ capsular fibroblasts,” it is not shown; please correct the flow gating.

2. The mean lifetime of interactions of CD4+ SP and CD8+ SP thymocytes with thymic antigen presenting cells during the induction of central tolerance is less than 1 hour (DOI: 10.1038/s41467-019-09727-4.). Positively selecting interactions are even more transient (DOI: 10.1126/scisignal.2004400). In Figure S3A, receiver MEFs were separated from sender MEFs at various time points, showing that mCherry was only reliably detected after 2 hours of contact with sender cells. Thus, the system does not seem to be sensitive enough to detect most interactions between self-reactive thymocytes and APCs presenting cognate self-antigens that drive central tolerance. This is a major limitation of the system that calls into question whether thymocyte: APC interactions of relevance to thymocyte selection can be observed with the YY system. On the one hand, the authors seem to acknowledge that cognate interactions cannot be detected in the limitations section, but on the other hand, they also claim based on blocking MHC-II-mediated interactions, that the detected interactions with thymocytes are mediated by MHC-II, which is contradictory. There is insufficient evidence that the receiver cells detected make physiologically relevant interactions with thymocytes.

3. On line 248-249, it is stated that, “DP and SP4 occupy distinct thymic anatomical locations, cortex and medulla, respectively. Hence, known anatomical locations occupied by senders and interacting cells facilitates assigning identified cellular interactors to their specific sender population.” However, DPs that have passed positive selection enter and accumulate in the medulla before downregulating CD8. Moreover, semi-mature CD4SP cells migrate within both cortex and medulla (DOI: 10.7554/elife.80443). Thus, it is not reasonable to assume that interactions occur in a given thymus location based solely on the thymocyte stage driving the interaction. Please modify the text throughout to remove this assumption, including in the next section on “characterization of the cellular interactome of SP4 and PD cells via flow cytometry....”

4. In Figure S3E, SP4, SP8, DP, Treg, CLP, and Gp38+ Fb are all indicated as receiver cells that made prolonged interactions with thymocytes from CD4-YY mice, in which most of the sender cells are CD4SP and DP. For all but CLP, these interactions were largely ablated upon treatment with a blocking antibody to MHCII. The authors state, “Supporting the biological relevance of the identified cellular-interactomes, treatment of CD4-YY mice with an aMHC-II blocking antibody (from 3 to 6 weeks of age) eliminated most mCherry+ cells from CD4-YY thymuses (Figs. 3E-F & S3E), proving that MHC-II mediates most of the identified interactions.” Thus, the authors interpret the reduction in labeled receiver cells by the blocking antibody to MHC-II to indicate that those receiver subsets underwent cognate interactions with CD4SP thymocytes. However, SP4, SP8, DP, and Treg do not express MHCII at appreciable levels. Thus, treatment with the anti-MHC-II antibody for 2 weeks likely had consequences beyond blocking cognate interactions, such as impaired positive selection of CD4SP thymocytes, which could have altered the thymus environment, resulting in the reduction in receiver subsets. Thus, the impact of findings from the YY system is diminished by 1) the previously reported transient nature of cellular interactions during positive and negative selection (see comment 2 above), with the much more prolonged interactions required to label receiver cells in the YY system, and 2) the lack of data supporting that labeled receiver subsets reflect cognate interactions with thymocytes. Altogether, although the YY system is elegant and recording cellular interactions in vivo is an important goal, it is not clear that the mCherry+ receiver cells in the YY reflect physiologically relevant cellular partners that promote thymocyte differentiation or selection. The lack of ability to detect interactions involved in antigen presentation with this system is acknowledged by the authors in the limitations section.

5. It will be important to distinguish whether DP sender cells in the CD4-YY mice are pre-positive selection, post-positive selection, or co-receptor reversing cells, or an equal mix of these subsets, as these are functionally distinct stages of T cell development with different localizations.

6. In lines 347-349, it is stated that “DCs also contribute to the development of self-tolerance by carrying peripheral antigens to the medulla”. Please broaden this statement to include discussion of relevant references showing the role of DCs in central tolerance to mTEC-derived self-antigens, endogenous self-antigens, as well as peripheral self-antigens (DOIs: 10.1038/s41577-024-01076-8, 10.1111/imr.70039). There is also relevant information in these references about the role of B cells in central tolerance that should be considered for the following sentence in the text.

Minor

1. In the introduction, please acknowledge previous approaches used to identify and quantify cell-cell interactions in the thymus using timelapse 2-photon imaging (DOIs: 10.1038/s41467-019-09727-4, 10.4049/jimmunol.1003563, 10.1038/ni.1761z)

2. The schematics are very helpful when evaluating the Yin/Yang system employed. However, in Figure 1A/B, it is not clear how the TRE is activated to drive mCherry expression in the receiver cells. Please modify the figure to indicate the location of tTA that activates TRE-mCherry. The explanation in the text on page 9 is very clear, but it will help the reader significantly to at least label the tTA domain in the receiver cells in Figure 1 Aii (right side).

3. In Supplementary Figure 1 Aii, some of the gels or primers seem to be mislabeled. They don't match the legend or expectations based on the schematics showing the locations of the primers.

4. In Figure 2D iv, given that a membrane-associated GFP was expressed, GFP should be localized to plasma membrane, as with the E2a-YY sender cells in figure 1Cii. But GFP does not appear to localized to the membrane of the thymocytes shown in Figure 2D iV, calling into question whether sender cells are truly visualized in the images.

5. If only Ubiq Cre-ERT2 data are shown in Figure 2D, then CD4-Cre should be eliminated to clarify.
6. What is the basis for gates in Figure 3E? It seems like the mCherry gate is too conservative, for example for DCs, and a much higher % are niche cells than shown.
7. Denoting CCR7+ DCs as migratory DCs (mig DC) is a misnomer in the thymus, as it has been shown that cDC1 differentiate from progenitors within the thymus and then become activated, while cDC2 migrate into the thymus in a non-activated state and then become activated within the sterile thymus environment (DOIs: 10.1016/j.immuni.2016.07.019, 10.4049/jimmunol.1700768, 10.1038/nr1385, 10.1084/jem.20082232, 10.1073/pnas.0810268105). Thus, please rename the "migDC" subset as "activated DCs" (aDCs).
8. In the Global Cell type dot plot in Figure S5B, please add at least one gene expressed by myeloid cells, as none are currently included to help evaluate annotation of this subset.

Reviewer #2

(Remarks to the Author)

In this manuscript, the authors developed a system for recording and reporting cell-cell interactions in the thymus using SynNotch, single cell sequencing, and CellPhoneDB. The combination of technologies is impressive and the experimental design is rigorous. The data indicate that the authors can successfully record cell-cell interactions in the thymus, a very complex tissue. This technology can likely also be applied to other organ systems.

Major Comments

1. The data in Figure 2Cii is n=2, which is insufficient to make conclusions. Additionally, the cumulative number of cell divisions is plotted. Since this is a cumulative number, it could only stay constant or increase. Somehow this number is decreasing in the first two panels, how is this possible?
2. The authors state that TM treatment induced 2.5% GFP sender cells (Fig. 2 D). Is this the highest level of sender cells that can be induced? Does adjusting the TAM does increase or decrease the number of sender cells? Or is this the highest possible number of sender cells generated by their system? It would be helpful to understand if there is any tunability in the system.
3. The manuscript could be strengthened if the authors could comment in the Introduction and/or Discussion on how these data and technologies could be used to enable development of new therapies.

Minor Comments

1. The alignment of the right panel of Figure 6A is slightly off, which makes it difficult to understand the ligand and receptor interactions.

Reviewer #3

(Remarks to the Author)

The article by Ganuza M. et al. employs a novel Yin & Yang (YY) system, based on a synNOTCH approach, to detect cellular interactomes at the single-cell level within the mouse thymus. The authors showed that the YY system can produce a durable expression of both mGFP and mCherry proteins beyond a week, and in vivo cellular interactions in the thymus can be observed through immunofluorescence, flow cytometry, and scRNAseq. With the YY system and different Cre-drivers, the authors were able to turn specific cell types into signal sender cells, and retain different cell types as receiver cells, allowing the detection of interactions between various cell types. Furthermore, mCherry+ recorded interactions by SP4 cells were analyzed using scRNAseq, capturing subsets of lymphocytes, stromal cells, and myeloid cells in the interactomes. Importantly, the detected interactions included mTECs, mFb, DCs, and B cells, which supports previous research on their contributions to SP4 differentiation. The analysis of two different age groups (8wks vs. 20wks) added another dimension to the significance of this study, showing a decline in interactions between SP4 and many cell types in the thymus. Molecular interactions in the cellular interactomes of SP4 and DP including Flt3/Flt3L, CD80/CD28, CD40/CD40L, Notch1/Jag1, and Ccr7/Ccl19 and Ccl21, are shown to be preserved from 8wks to 20wks in mouse thymus. Taken together, the findings are clearly presented, rigorously supported, and establish a novel and powerful system for studying the cellular interactomes between developing T cells and relevant thymic niche cells.

Potential minor revisions:

1. In Fig.6A, it might have been informative to separate the DN_DP(P) group into DN and DP will clarify and provide more information regarding the interactions of each subset with the niche cells. For example, how does Notch signalling influence DN and DP cells differently, and which niche cells are responsible for these interactions at each stage?
2. In Fig.2B, it seems that the gating of mCherry-High cells may be a bit too generous. Some mCherry-Low cells are also included, and could affect the downstream analysis.
3. In Fig.3E, the expression of mCherry rises beautifully in receiver cells when they interact with the senders. However, their mGFP level also rises, which is a bit confusing and the authors may want to address this concern.

Editing/clarifications:

1. In Line 145, the authors could better define white blood cells by explaining the previous gating approach in their flow cytometric analysis? Was it only gated on FSC and SSC?
2. In Line 371-372, I would suggest using the following expression: "Thymuses from 20-weeks old.....**served as controls** for assessing the enrichment in mCherry+ cells...."
3. In the future, as part follow up work, it would be interesting to see comparisons between aged groups that are farther apart — for example, 1yr old vs. 8wks old mice.
4. In Line 422-423, I would suggest using the following expression: "Several components of the.....produced TGFb, **which is** crucial for SP4 differentiation to Tregs."
5. In Line 423-426: "Pro-survival signalling as those **triggered** by.....mediated by **DCs** and B cells...."

Version 1:

Reviewer comments:

Reviewer #1

(Remarks to the Author)

The authors have carried out a thorough experimental and text revision that clearly and concisely addresses previous comments. In particular, new data in Figure S3B showing that 3-6 sequential transient interactions of 30' each is sufficient to induce mCherry expression in receiver cells demonstrates that the interactions recorded in the thymus microenvironment have the potential to reflect cognate interactions inducing positive or negative selection. These data support the relevance of the YinYang system to studying key cellular interactions driving selection events in situ. Moreover, the revised gating schemes in S2A and improve the accuracy of the cell types quantified as niche/receiver cells. Altogether, the authors have generated and carefully characterized an elegant system for studying cellular interactions during thymocyte differentiation and selection. I congratulate them on this important new model for the field and anticipate that many new insights will come from it.

There are two revisions needed (no experiments needed) prior to publishing to improve accuracy of data and interpretations presented:

1. It is greatly appreciated that the authors have added flow cytometry gating for DP subsets to delineate which are senders, and it's notable that in the CD4-YY system, mainly post-positive selection DP subsets are senders (Figure 3Cii). However, the DP gating scheme needs to be revised because the non-sequential scheme in S2v results in overlap between subsets and thus, inaccurate quantification for Figure 3Cii.

The gated "TCRab^{high} CD69+CD5+CCR7+ CCR4-" DP cells are co-receptor reversing DPs, although they are currently labeled as "CCR4- post positive selection DPs". These cells will overlap with the population currently labeled as "co-receptor reversal DP" because of the lack of sequential gating (both subsets are DP TCRab^{high}CCR7+ cells, as gated). I suggest not using additional restrictions on the CD4+CD8+ DP gate when identifying co-receptor reversing DPs, as these cells are in flux from CD4SP semi-mature cells and move through different levels of CD4 and CD8 as they fully differentiate to become CD8SPs. The co-receptor reversing DPs can be distinguished by expression of CD69 and high levels of CD3 or TCRab.

Also, In S2v, you've currently gated out most of the early post-positive selection DPs, as these are the TCRab low cells between your gates in the DAPI vs TCRab flow cytometry plot on the top row: to gate on these early post-positive selection DPs, you can gate CD69+ CD3 lo, or CD69+ TCRab low, cells within the DP gate, and you'll find that the majority (80-90%) express CCR4 but not CCR7. It's great that you have both of these chemokine receptor markers in your stain already.

To simplify and more accurately gate on non-overlapping DP subsets, I suggest drawing three gates to subdivide DPs, using a flow cytometry plot of CD69 vs either CD3 or TCRab: 1) CD69- and CD3lo or TCRab^{lo} DPs are the pre-selection cells (they'll express neither CCR4 nor CCR7); 2) CD69+ CD3lo or TCRab^{lo} DPs are early post-selection DPs (which are almost all CCR4+ CCR7-); 3) : CD69+ CD3hi or TCRab^{hi} are co-receptor reversing cells—this is a distinct population with good CD3 and CD69 resolution— (many of these express both CCR4 and CCR7, but there are some that express only one of these CCRs). With these 3 gates, you can easily distinguish three stages of DPs (pre-selection, post-selection, and co-receptor reversing) without any redundancy between them for accurate quantification.

2. The previous Major Concern 4 was raised because DPs, CD4SPs, and CD8SPs were identified as receivers (of DP and CD4SP interactions) and their mCherry expression was reduced after MHC-II blockade, despite the fact that these receiver subsets don't express MHCII in mice. The authors have now added a thoughtful discussion (Lines 283-315) suggesting that prolonged MHCII blockade could result in secondary consequences to the thymus environment that reduce non-cognate interactions of with these thymocyte receiver cell types. However, they have retained the possibility that reduced mCherry in these subsets could indicate a reduction in MHCII-mediated thymocyte-thymocyte interactions, stating "Furthermore, thymocyte-thymocyte interactions have been shown to be important for "efficient positive selection and maturation of SP4 cells" [PMID: 16226504, PMID: 11292248, PMID: 21478404, PMID: 23264931, PMID: 20038602, PMID: 23776174], further supporting the biological significance of the interactions identified in this study." This sentence should be removed, as the cited studies do not support that wild-type thymocyte-thymocyte interactions in mice induce positive selection on MHCII. Choi EY et al., Immunity 23, 387-396 (2005) PMID: 16226504, and the other cited papers from this research group, use an LCK promoter-driven Ciita transgene to ectopically express MHCII on thymocytes. Mouse thymocytes do not otherwise express MHCII. Through ectopic MHCII expression, the authors demonstrate that mouse thymocytes CAN induce positive

selection. If you look at the bone marrow chimeras in Figure 3 of PMID: 16226504, you'll see that WT BM transferred into Ciita^{-/-} recipients does not result in positive selection of CD4SP cells, demonstrating there is no natural thymocyte-thymocyte mediated positive selection on MHCII. Because mouse thymocytes don't naturally express MHCII, in the current study, the reduced interactions of CD4SP with thymocyte subsets strongly implicates a secondary effect of MHCII blockade on the thymus environment, as opposed to a direct effect of interfering with cognate interactions. The idea, however, that thymocyte-thymocyte interactions induce agonist selection of innate thymocyte subsets is well supported by the literature (but this isn't driven by MHCII-mediated interactions naturally, although the ciita transgenic model led to important insights about innate lymphocyte selection). Also, human thymocytes express MHCII, so they may contribute to positive selection of CD4SP, but that it is not relevant for results in this mouse model.

Reviewer #2

(Remarks to the Author)

The authors have addressed my previous concerns and I have no further comments. The technologies reported in this manuscript will have a broad impact for mapping cellular interactomes in vivo, and the reported data provide new insights into thymus biology.

Reviewer #3

(Remarks to the Author)

The authors have fully and carefully addressed all my initial concerns, and should be congratulated on their exciting new mouse model system and for the insights derived from their analysis.

Sánchez-Lanzas *et al.* response to Reviewers:

We thank the Reviewers for their constructive and positive comments, which have helped us improve our manuscript. All changes are highlighted in yellow in the manuscript, and our specific responses are detailed below, point-by-point. We first provide a summary of the major changes.

Summary of major changes/responses

- **The methodology by which we identify thymic populations by flow cytometry has been extensively revised and updated (in response to helpful suggestions from Reviewer-1, and Reviewer-3).**
- **We provide strong new experimental evidence that our *Yin&Yang* system is able to record brief repetitive intermittent interactions, e.g. three repetitive 30-min interactions, that more faithfully represent those interactions in biological systems (e.g. selection events in the thymus).**
- **We address in detail the consequences of blocking MHC-II in the thymus. In particular, we embrace the additional biological insights revealed for cellular interactions that may be indirectly disrupted when certain other cellular interactions are directly perturbed.**
- **We discuss the concept of physiologically relevant interactions from the perspective of the development of the *Yin&Yang* system. Specifically, that our new tool reveals, in an unbiased manner, cellular interactions that exist within the operational parameters of the system, that we have carefully defined. That the biological relevance of some of these interactions may not yet be known is, in our opinion, a great advantage, allowing researchers to further explore interactions that *heretofore* may not have been predicted.**

Reviewer #1 – Comments (R1C):

R1C-1a (Major-1): Because identification of sender and receiver cells relies on accurate flow cytometry gating, it is critical that the cell populations in the thymus are more accurately defined by flow cytometry. Figure S2Ai shows a gating strategy to identify LMPP, CLP, and ETP in the thymus. However, LMPP and CLP are present in bone marrow, not thymus, while ETP are present in the thymus. So, it is not clear what populations or organs are being analyzed, as the legend indicates that the cells shown are from the thymus. It's possible that the Flt3⁺ cells are DC precursors in the thymus (doi 10.1002/eji.201141728). ETP are typically defined as lineage- cKIT⁺ CD44⁺CD25⁻, as later acknowledged by the authors, with the most immature thymocyte progenitors expressing Flt3. The LMPP gate and CLP gate do not look clean based on CD127 staining (if this is a BM analysis, which I'm not sure it is). Please gate the populations more precisely in the thymus, which will be critical for proper analysis of sender and receiver cells.

R1C-1b: In S2Aii, Treg should be defined as CD4SP thymocytes (CD3⁺) that express CD25 and Foxp3. The gate on CD25 does not clearly delineate the Treg population.

R1C-1c: In S2 A iii, please correct the x-axis in the 2nd flow cytometry panel (CD11c); however, DCs in the thymus should be gated as CD11c⁺MHCII⁺ cells that do not express F4/80 to avoid macrophages. Also, Gp38/podoplanin is expressed by both mesenchymal cells and TECs in the thymus,

so the CD45-Gp38⁺ cells will be a mix of these cell types. If a CD26 gate is used to define the “CD26⁺ capsular fibroblasts,” it is not shown; please correct the flow gating.

Our response: We thank the Reviewer for these helpful comments. We have now redefined the populations as suggested by the Reviewer (see below) and have updated the references accordingly and gating strategies in Figure SF2A and in the Methods section (“Thymus analysis” lines 674-716).

ETPs: Are re-gated now as Lin⁻(CD11b⁻CD19⁻NK1.1⁻Ter119⁻) cKit⁺CD44⁺CD25⁻ [PMID: 15851035; PMID: 7655014; PMID: 17582341]. Updated in Line 272 and 679.

CD11c⁺ cells: Are now described as “CD45⁺CD11c⁺ containing ~75% of DCs and ~25% of macrophages” [PMID: 36449334; PMID: 19273629] in lines 270-271 and 691-692. Figures 3 & 5, SF4 have been edited accordingly with legends indicating CD11c⁺ cells instead of DCs.

CD25⁺ SP4 T cells: Are now described as “CD19⁻CD11b⁻NK1.1⁻TER119⁻CD8⁻CD4⁺CD25^{high} enriched for T regulatory cells [PMID: 11564442]” in lines 682-683. Figures 5Bi, SF2Aii, SF3F, SF4A, SF4B have been updated accordingly.

CD45⁺FIt3⁺CD4⁻CD8⁻CD25⁻ cells (containing DC precursors): Is now included as suggested [PMID: 21630253] in lines 271-272 and 680-681. Analyses related to these cells have been included in Figures 5Ai, SF3 and SF4, and cells are now identified as FIt3⁺.

Medullary fibroblasts: Are now described as CD45⁻CD31⁻EPCAM⁻Gp38⁺CD26⁻ [PMID: 35587733; PMID: 39112630; PMID: 36413155] in lines 692-693. Figures SF2Aiii, SF3F and SF4Bii have been edited accordingly.

Capsular fibroblasts: Are now described as CD45⁻CD31⁻EPCAM⁻CD26⁺ [PMID: 35587733; PMID: 39112630; PMID: 36413155] in lines 273-274 and 693-694. Figures SF2Aiii, SF3F and SF4Bii have been updated.

Cortical TECs: Are now described as CD45⁻CD31⁻EPCAM⁺UEA-1⁻Ly51⁺(62-64, 135) [PMID: 35559672; PMID: 35587733; PMID: 25973789; PMID: 11792372] in line 696. Figures SF2Aiii, SF3F and SF4Bii have been edited accordingly.

LMPPs and CLPs: Were removed.

Gating strategy for CD26⁺ capsular fibroblasts: (CD45⁻CD31⁻EPCAM⁻CD26⁺) is now shown in SF2Aiii.

X-axis in SF2Aiii: This has now been corrected from CD11 to CD11c.

In brief, we have updated our gating strategies and the descriptions of these populations in the text to ensure that they can be properly understood by the reader.

RIC-2 (Major-2): *The mean lifetime of interactions of CD4⁺ SP and CD8⁺ SP thymocytes with thymic antigen presenting cells during the induction of central tolerance is less than 1 hour (DOI: 10.1038/s41467-019-09727-4.). Positively selecting interactions are even more transient (DOI: 10.1126/scisignal.2004400). In Figure S3A, receiver MEFs were separated from sender MEFs at various time points, showing that mCherry was only reliably detected after 2 hours of contact with sender cells. Thus, the system does not seem to be sensitive enough to detect most interactions between self-reactive thymocytes and APCs presenting cognate self-antigens that drive central tolerance. This is a major limitation of the system that calls into question whether thymocyte: APC interactions of relevance to thymocyte selection can be observed with the YY system. On the one hand, the authors*

seem to acknowledge that cognate interactions cannot be detected in the limitations section, but on the other hand, they also claim based on blocking MHC-II-mediated interactions, that the detected interactions with thymocytes are mediated by MHC-II, which is contradictory. There is insufficient evidence that the receiver cells detected make physiologically relevant interactions with thymocytes.

Our response: We thank the Reviewer for highlighting this important point related to the sensitivity of the *Yin&Yang* system to detect brief interactions (<1hr), such as those involved in selection events in the thymus.

Following cellular interactions, the recorded signal (H2BmCherry) is very stable in our *Yin&Yang* system: at least six days and five cell divisions (Figure 2C). Thus, intermittent (even short) interactions are expected to progressively build up the intensity of H2BmCherry to detectable levels. To demonstrate this, now **we provide new data** showing that three repetitive intermittent interactions of 30-mins lead to detectable H2BmCherry levels (New Supplemental Figure S3B). Thus, we believe our system is capable of detecting brief intermittent interactions, such as those involved in selection events in the thymus.

We have included text pertaining to this in the manuscript as follows:

Lines 175-177: “mCherry-based recording of cell-interactions reports brief intermittent repetitive contacts (<1 hour) and is detectable for at least 6 days and 5 cellular divisions post-encounter.”

Lines 188-189: “Moreover, three repetitive intermittent interactions of 30 minutes were enough to yield detectable mCherry induction (Fig. S3B).”

Lines 198-201: “Overall, these data demonstrate that the high stability of the recorded signal (*i.e.* H2BmCherry) enables to record transient cellular interactions, such as those involved in the induction of central tolerance and positive selection, which have been reported to last less than one hour [PMID: 24129702; PMID: 31101805]”

Lines 541-546: “Importantly, our system is capable of detecting brief cellular interactions (< 1hour) (Fig. S3B) such as those involved in the induction of central tolerance and positive selection [PMID: 24129702; PMID: 31101805]. Notably, the H2B-mCherry-based interaction-recording system exhibits high stability (persisting for at least six days and five cell divisions, Fig. 2C), which undoubtedly facilitates the detection of such transient events. Indeed, even three intermittent interactions of 30 minutes each were sufficient to generate detectable mCherry signals (Fig. S3B).”

Supplemental Figure 3B. “Three repetitive intermittent cell interactions of 30 minutes result in detectable recording of cell interactions. Interacting receiver cells get labeled as H2B-mCherry⁺. **Bi.** Experimental schematic. Senders MEFs (grown on cover slips) were put in contact with receiver MEFs for 30 mins. Coverslips enabled timely removal of receivers. After a one-hour non-interacting gap, interaction was repeated or not. The 30 minutes interaction + 1hour gap was repeated x1, x3 or 6x times. Receiver cells were FACS-sorted to eliminate any sender that could have been transferred to the coverslip. Sorted receivers were cultured for additional 36 hours to enable mCherry expression. **Bii.** Representative flow cytometry plots of sorted receivers 36 hours post-culture. **Biii.** Quantification of mCherry⁺ induction following repetitive intermittent interactions. (n=7 biological replicates, results pooled from two independent experiments.”

Lines 515-518: “Consistent with the interdependence of thymic cell populations in maintaining homeostasis [PMID: 20859118; PMID: 32082299], global MHC-II blockade broadly disrupted thymic interactions, underscoring our system’s ability to detect both known and previously unrecognized cellular connections.”

Lines 561-568: “Additionally, LIPSTIC only allows tracking cell-interactions for short periods of time (substrate labeling is detectable only for 4-6 hours post-administration)(118). By contrast, H2B-mCherry-based recording of cell interactions in our YY mice is stable (6 days and 5 cell divisions), allowing for long-term tracking of cellular interactions (Fig. 2C). Moreover, our system enables detection of repetitive brief interactions shorter than one hour (Fig. S3B). Both mouse models could potentially be used simultaneously as they both depend on CRE expression to induce senders (in YY mice) and substrate donors (in LIPSTIC mice) complementing each other.”

We have also amended the “limitations” section of the manuscript (previously in discussion, lines 528-530) and removed associated limitations: ~~Our YY system is biased to detect “stable” cellular interactions. This precludes to record brief interactions important in antigen presentation in the thymus.~~

RIC-3 (Major-3): *On line 248-249, it is stated that, “DP and SP4 occupy distinct thymic anatomical locations, cortex and medulla, respectively. Hence, known anatomical locations occupied by senders and interacting cells facilitates assigning identified cellular interactors to their specific sender population.” However, DPs that have passed positive selection enter and accumulate in the medulla before downregulating CD8. Moreover, semi-mature CD4SP cells migrate within both cortex and medulla (DOI: 10.7554/elife.80443). Thus, it is not reasonable to assume that interactions occur in a given thymus location based solely on the thymocyte stage driving the interaction. Please modify the text throughout to remove this assumption, including in the next section on “characterization of the cellular interactome of SP4 and PD cells via flow cytometry....”*

Our response: We thank the Reviewer for these important comments and have amended the text accordingly by removing assumptions based on the location of DP and SP4 to assign cell interactions, and any related discussion. The following sections have been removed:

Previously in Lines 99-101 (Introduction): ~~“Known anatomical locations occupied by DP and SP4 cells (cortex and medulla, respectively) and those by interacting cell types helped to dissect out cellular interactors”.~~

Previously in Lines 248-251 (Results): ~~“DP and SP4 occupy distinct thymic anatomical locations, cortex and medulla, respectively. Hence, known anatomical locations occupied by senders and interacting cells facilitates assigning identified cellular interactors to their specific sender population.”~~

Discussion – Previously in Lines 447-449: ~~“Previously known anatomical locations of the identified interacting cells helped us to discern which among those related to the interactomes of medullary SP4 and cortical DP cells.~~

Lines 275-277: We have removed comments suggesting that we can assign interactions to specific senders based on anatomical locations and we just indicate where the cellular interactors are more frequently located. “Most of these cell interactors (except for ETPs and capsular Fb) concentrate in the medulla. Meanwhile, thymic eosinophils congregate in the

corticomedullary junction and the medulla, and less frequently in the cortex and interlobular septa.”.

In lines 378-379: we indicate now that “Most detected cellular interactors are known to anatomically locate preferentially to the medulla”

And in lines 398-399: we indicate that, “A subset of identified cellular interactors known to localize preferentially near or within the thymic cortex was also detected in our analysis”.

R1C-4 (Major-4): *In Figure S3E, SP4, SP8, DP, Treg, CLP, and Gp38+ Fb are all indicated as receiver cells that made prolonged interactions with thymocytes from CD4-YY mice, in which most of the sender cells are CD4SP and DP. For all but CLP, these interactions were largely ablated upon treatment with a blocking antibody to MHCII. The authors state, “Supporting the biological relevance of the identified cellular-interactomes, treatment of CD4-YY mice with an aMHC-II blocking antibody (from 3 to 6 weeks of age) eliminated most mCherry+ cells from CD4-YY thymuses (Figs. 3E-F & S3E), proving that MHC-II mediates most of the identified interactions.” Thus, the authors interpret the reduction in labeled receiver cells by the blocking antibody to MHC-II to indicate that those receiver subsets underwent cognate interactions with CD4SP thymocytes. However, SP4, SP8, DP, and Treg do not express MHCII at appreciable levels. Thus, treatment with the anti-MHC-II antibody for 2 weeks likely had consequences beyond blocking cognate interactions, such as impaired positive selection of CD4SP thymocytes, which could have altered the thymus environment, resulting in the reduction in receiver subsets. Thus, the impact of findings from the YY system is diminished by 1) the previously reported transient nature of cellular interactions during positive and negative selection (see comment 2 above), with the much more prolonged interactions required to label receiver cells in the YY system, and 2) the lack of data supporting that labeled receiver subsets reflect cognate interactions with thymocytes. Altogether, although the YY system is elegant and recording cellular interactions in vivo is an important goal, it is not clear that the mCherry+ receiver cells in the YY reflect physiologically relevant cellular partners that promote thymocyte differentiation or selection. The lack of ability to detect interactions involved in antigen presentation with this system is acknowledged by the authors in the limitations section.*

Our response: These are important considerations. In our reply to point R1C-2 above, we demonstrate with additional data the ability of the Yin&Yang system to detect repetitive intermittent short interactions aided by the stability of the recorded signal (H2BmCherry). This shows that our system can detect brief interactions lasting much less than an hour, such as in the biologically relevant cognate interactions involved in antigen presentation. Treatment with an anti-MHC-II antibody blocked these biologically relevant cognate interactions, as expected (Fig 3E-F).

However, as the Reviewer astutely points out, we also observed that anti-MHC-II treatment also blocked other interactions (e.g. with SP4, SP8, DP). We agree with the Reviewer that these effects are likely indirect, being the consequence of perturbations to the thymic environment as a result of blocking MHC-II interactions with cognate receptors such as the TCR. However, we don't believe that these considerations undermine the biological relevance of those interactions. On the contrary, we welcome the additional biological insights revealed for cellular interactions that may be indirectly disrupted when certain other cellular interactions are directly perturbed. Indeed, this highlights the capacity of the Yin&Yang system to identify interactions that may not yet be recognised.

Finally, it is worth also considering that MHC-II-mediated thymocyte-thymocyte interactions in the thymus have been reported, particularly in the context of innate T cell development:

Choi EY et al., *Immunity* 23, 387-396 (2005) PMID: 16226504

Choi, EY. et al, *Cell Mol Biol* 47, 135-143 (2001) PMID: 11292248

Min HS et al., *J Immunol* 186, 5749-5757 (2011) PMID: 21478404

Qiao Y et al., *Open J Immunol* 2, 25-39 (2012) PMID: 23264931

Lee YJ et al., *J Exp Med* 207, 237-246 (2010). PMID: 20038602

Zhu L et al., *J Immunol* 191, 737-744 (2013). PMID: 23776174

Thus, we contend that, within its well-characterised operational parameters, the Yin&Yang system has the capacity to report real cellular interactions that are physiological in nature. The significance, relevance and importance of these interactions may not yet be evident, but as an hypothesis-generating discovery tool, the Yin&Yang system can reveal cellular interactions that heretofore may not have been predicted.

We discuss these considerations in the sections below:

Lines 283 – 315: “Supporting the biological relevance of the identified cellular-interactomes, treatment of *CD4-YY* mice with an α MHC-II blocking antibody (administered from 3 to 6 weeks of age) eliminated most mCherry⁺ cells from *CD4-YY* thymuses (Figs. 3E-F & S3F). These cells included those engaged in well-established interactions between SP4/DPs and APCs, such as DCs and B cells mediated by the MHC-II – TCR molecular axis [PMID: 33519827; PMID: 26070482; PMID: 40625741]. Intriguingly, interactions involving other cell types within the SP4/DP interactome that exhibit low or negligible MHC-II expression [PMID: 23441229; PMID: 17125142], including SP4, SP8, ISP, DP (74), were also reduced following α MHC-II antibody treatment, which was statistically significant for ISPs ($p < 0.05$) but did not reach statistical significance in SP4, SP8 and DPs (all $p < 0.1$) (Fig. S3E).

Even though the effect of the α MHC-II blocking antibody on interactions with non-expressing MHC-II cells was, as expected, more subtle, the general reduction in interacting cells suggests that α MHC-II treatment may induce broader perturbations beyond the direct blockade of cognate MHC-II–TCR interactions. Such perturbations may include impaired positive selection of SP4 thymocytes, which could, in turn, disrupt the thymic microenvironment and lead to a reduction in downstream cellular subsets.

These findings align with the well-established interdependence among multiple thymic cell populations required to maintain thymic homeostasis [PMID: 20859118; PMID: 32082299]. For example, the maintenance of both cTECs and mTECs in adulthood depends on the presence of functional thymocytes [PMID: 20859118; PMID: 16274965]. Indeed, “thymuses in which T-cell development is arrested at the DN1 stage exhibit a complete loss of cTECs and mTECs” [PMID: 16274965], a phenotype that can be rescued by the introduction of functional T-cell progenitors [PMID: 15265888]. This demonstrates that, similar to thymocytes, TECs undergo continuous turnover [PMID: 16896157]. Conversely, a deficiency of mature TECs profoundly reduces overall thymic cellularity, thereby preventing the development of $\alpha\beta$ T cells and invariant NK cells [PMID: 25770130].

Thus, it is unsurprising that global MHC-II blockade exerts widespread effects on thymic cellular interactions. More importantly, these results highlight the ability of our system to capture

diverse types of cellular interactions, revealing both established and previously unrecognized connections that may open new avenues of investigation. Furthermore, thymocyte–thymocyte interactions have been shown to be important for “efficient positive selection and maturation of SP4 cells” [PMID: 16226504, PMID: 11292248, PMID: 21478404, PMID: 23264931, PMID: 20038602, PMID: 23776174], further supporting the biological significance of the interactions identified in this study.”

Lines 541-546: “Importantly, our system is capable of detecting brief cellular interactions (< 1hour) (Fig. S3B) such as those involved in the induction of central tolerance and positive selection [PMID: 24129702; PMID: 31101805]. Notably, the H2B-mCherry-based interaction-recording system exhibits high stability (persisting for at least six days and five cell divisions, Fig. 2C), which undoubtedly facilitates the detection of such transient events. Indeed, even three intermittent interactions of 30 minutes each were sufficient to generate detectable mCherry signals (Fig. S3B)”.

RIC-5 (Major-5): *It will be important to distinguish whether DP sender cells in the CD4-YY mice are pre-positive selection, post-positive selection, or co-receptor reversing cells, or an equal mix of these subsets, as these are functionally distinct stages of T cell development with different localizations.*

Our response: We have now further characterised DP sender subsets as suggested. We included these results in Figure 3Cii and reported on this in the text in lines 248-253 and 256-258 as follows:

Lines 248-253: “Further analysis on the identity of DP senders (31, 50-55) revealed that 44% of them were CCR4⁻ post-positive selection DPs (TCRαβ^{HIGH}CD69⁺CD5⁺CD62L⁻CCR7⁺), 2% were CCR4⁺ post-positive selection DPs and 21% were co-receptor reversal DPs (TCRαβ^{HIGH}CD4⁺CD8^{-/low} CCR7⁺) (Fig. 3Cii). Pre-positive selection DP (TCRαβ^{-/low}CD69⁻CD5⁻CD62⁻CCR7⁻) senders were not detected (Fig. 3Cii).”

Lines 256-258: “However, given that 30% of the senders were DPs in CD4-YY thymuses, ~30% of the detected interactions would be driven by DPs (mainly post-positive selection DPs)”.

RIC-6 (Major-6): *In lines 347-349, it is stated that “DCs also contribute to the development of self-tolerance by carrying peripheral antigens to the medulla”. Please broaden this statement to include discussion of relevant references showing the role of DCs in central tolerance to mTEC-derived self-antigens, endogenous self-antigens, as well as peripheral self-antigens (DOIs: 10.1038/s41577-024-01076-8, 10.1111/imr.70039). There is also relevant information in these references about the role of B cells in central tolerance that should be considered for the following sentence in the text.*

Our response: As suggested by the Reviewer we have broadened this statement on the role of DCs and B cells. This is indicated in the following text:

Lines 383-403: “DCs play a crucial role in establishing central tolerance by presenting self-antigens derived from diverse sources [PMID: 33519827; PMID: 40433811; PMID: 19935803]. This includes mTECs-derived self-antigens, which are essential for promoting tolerance to antigens expressed in a limited subset of peripheral tissues. These antigens are presented by activated DC1 (aDC1) and aDC2 subsets. In parallel, peripheral self-antigens carried into the thymic medulla by cDC2 cells enable tolerance to self-antigens that are not expressed by mTECs [PMID: 31101805; PMID: 29752065; PMID: 35099391; PMID: 25220213; PMID: 37067332].”

Specialized subsets of aDC2s (specifically CCR2⁺ and CX3CR1⁺ populations) transport and present serum-borne circulating antigens [PMID: 19675159; PMID: 34711828]. The role of pDCs in the establishment of central tolerance is less well characterized [PMID: 22444632]. Likewise, B cells contribute to central tolerance through the presentation of self-antigens [PMID: 33519827; PMID: 26070482; PMID: 20976010; PMID: 25344473; PMID: 10753841; PMID: 24082098]. Upon entry into the thymus, B cells upregulate MHC-II and CD80 expression in a CD40 signaling-dependent manner. Unlike their peripheral counterparts, thymic B cells also express Aire [PMID: 26070482; PMID: 20976010; PMID: 25344473; PMID: 10753841; PMID: 24082098]. Moreover, B cell receptor (BCR)-mediated antigen uptake and presentation has been also identified as an additional mechanism facilitating central tolerance [PMID: 33446417; PMID: 31591259]. In addition, thymic B cells can promote Treg induction [PMID: 26070482; PMID: 26070482].

A subset of identified cellular interactors known to localize preferentially near or within the thymic cortex [PMID: 33519827; PMID: 28530714] was also detected in our analysis. These included capsular perilobular fibroblasts, ETPs and iELpA (Fig. 4B). While the role of iELpA in thymocyte development remains unknown [PMID: 32079746; PMID: 28530714], capsular fibroblasts provide structural support within the thymus. The interplay between these fibroblasts and cTECs may contribute to T-cell development, although this relationship has yet to be fully elucidated [PMID: 34096078].

RIC-7 (Minor-1): *In the introduction, please acknowledge previous approaches used to identify and quantify cell-cell interactions in the thymus using timelapse 2-photon imaging (DOIs: 10.1038/s41467-019-09727-4, 10.4049/jimmunol.1003563, 10.1038/ni.1761z).*

Our response: We thank the Reviewer for highlighting this important missing information. We have now included it in:

Lines 60-61: “including timelapse 2-photon and 3-photon imaging that have allowed the identification and quantification of cell-cell interactions [PMID: 31101805; PMID: 41168496; PMID: 21278345; PMID: 19066556; PMID: 19543275].”

RIC-8 (Minor-2): *The schematics are very helpful when evaluating the Yin/Yang system employed. However, in Figure 1A/B, it is not clear how the TRE is activated to drive mCherry expression in the receiver cells. Please modify the figure to indicate the location of tTA that activates TRE-mCherry. The explanation in the text on page 9 is very clear, but it will help the reader significantly to at least label the tTA domain in the receiver cells in Figure 1Aii (right side).*

Our response: We have now amended Figure 1Aii to include a schematic indicating the location of the tTA in the GFP receptor. Text in Figure legend 1Aii (lines 1272-1275) was added as follows:

“The GFP- REC (a synthetic NOTCH receptor) is formed by an anti-GFP nanobody (located outside of the cell membrane) bound to a MYC-TAG, a NOTCH transmembrane domain (TMB) and a cytoplasmic tetracycline transactivator (tTA). Schematic is shown on the right”.

Additionally, we added a comment in Figure legend 1Aii (line 1286) prompting the reader to Figure 2A for further details in the mechanism of activation as follows:

“(Details on the mechanism of activation shown in Figure 2A).”

RIC-9 (Minor-3): In Supplementary Figure 1 Aii, some of the gels or primers seem to be mislabeled. They don't match the legend or expectations based on the schematics showing the locations of the primers.

Our response: We thank the Reviewer for identifying this typo. The error comes from mislabelling $\gamma\gamma^{mGFP/REC}$ mice as $E2\alpha^{+/CRE} \gamma\gamma^{mGFP/mGFP}$. $\gamma\gamma^{mGFP/REC}$ mice carry both $\gamma\gamma^{Rec}$ and $\gamma\gamma^{mGFP}$ alleles. We have edited SF1Aii and text in SF1Aii figure legend accordingly:

“ $\gamma\gamma^{mGFP/REC}$ mice (carrying both alleles)”

RIC-10 (Minor-4): In Figure 2D iv, given that a membrane-associated GFP was expressed, GFP should be localized to plasma membrane, as with the E2a-YY sender cells in figure 1Cii. But GFP does not appear to localize to the membrane of the thymocytes shown in Figure 2D iV, calling into question whether sender cells are truly visualized in the images.

Our response: We have now provided improved new images for this Figure.

RIC-11 (Minor-5): If only Ubiq Cre-ERT2 data are shown in Figure 2D, then CD4-Cre should be eliminated to clarify.

Our response: As suggested, we removed CD4-Cre from Figure 2D for clarity.

RIC-12 (Minor-6): What is the basis for gates in Figure 3E? It seems like the mCherry gate is too conservative, for example for DCs, and a much higher % are niche cells than shown.

Our response: Gates for mCherry have been adjusted in Figure 3E and all related graphs amended in Figures 3F, 5B and SF3F.

RIC-13 (Minor-7): Denoting CCR7+ DCs as migratory DCs (mig DC) is a misnomer in the thymus, as it has been shown that cDC1 differentiate from progenitors within the thymus and then become activated, while cDC2 migrate into the thymus in a non-activated state and then become activated within the sterile thymus environment (DOIs: 10.1016/j.immuni.2016.07.019, 10.4049/jimmunol.1700768, 10.1038/ni1385, 10.1084/jem.20082232, 10.1073/pnas.0810268105). Thus, please rename the “migDC” subset as “activated DCs” (aDCs).

Our response: We thank the Reviewer for this very insightful comment. We have renamed migDC as activated (aDCs) across the text in:

Lines 354 [“Ccr7⁺Lamp3⁺ activated DCs (aDCs)”], **Line 366** (“aDCs”), **Line 477** (“aDCs”) and in all related Figures (Fig 5Aii, 5Bii, 6, SF5B, SF6A, SF6C)

RIC-14 (Minor-8): In the Global Cell type dot plot in Figure S5B, please add at least one gene expressed by myeloid cells, as none are currently included to help evaluate annotation of this subset.

Our response: We have now included gene expression for *Cd14*, *Cd16 (Fcgr3)* (monocytes), *Itgam*, *Mrc1* (macrophages) and *H2-Aa* (DCs) used to identify myeloid cells in the Global Cell type plot in Fig S5B.

Reviewer #2 – Comments (R2C):

R2C-1 (Major-1): *The data in Figure 2Cii is n=2, which is insufficient to make conclusions. Additionally, the cumulative number of cell divisions is plotted. Since this is a cumulative number, it could only stay constant or increase. Somehow this number is decreasing in the first two panels, how is this possible?*

Our response: We thank the Reviewer for spotting this issue. The Reviewer is correct that the number of cell divisions can only increase. We have now repeated these experiments (n=4 from 3 independent experiments) and reanalysed data. This allowed us to statistically determine that cell contact recording lasts for at least 6 days post-contact and 5 cell divisions. This is now included in the text in lines: 176, 196, 564.

R2C-2 (Major-2): *The authors state that TM treatment induced 2.5% GFP sender cells (Fig. 2 D). Is this the highest level of sender cells that can be induced? Does adjusting the TAM does increase or decrease the number of sender cells? Or is this the highest possible number of sender cells generated by their system? It would be helpful to understand if there is any tunability in the system.*

Our response: We thank the Reviewer for this very relevant comment. Ubiq-CreERT2 mice could be treated longer. In the depicted experiments mice were treated with tamoxifen for 11-days. Thus, we expect that similar to other floxed alleles (as we reported in Sanchez-Lanzas et al., Leukemia 2025 in Figure 1D, PMID: 39537978), longer exposure with TAM will increase recombination efficiency. We commented on this in the text in:

Lines 580-582: “Accordingly, the efficiency of recombination driven by any other CRE line on producing sender cells will need to be evaluated and optimized. For instance, longer tamoxifen exposures are expected to increase recombination efficiency in CRE-ERT2 lines [PMID: 39537978].”

R2C-3 (Major-3): *The manuscript could be strengthened if the authors could comment in the Introduction and/or Discussion on how these data and technologies could be used to enable development of new therapies.*

Our response: We thank the Reviewer for this suggestion. We have now commented in the discussion on how to employ these technologies to identify potential therapeutical targets in:

Lines 522-532: “Importantly, the observed reduction in the frequency of detected interactions during thymic involution in 20 weeks old mice (Fig. 5B) provides an additional layer of complexity to our understanding of how ageing progressively erodes the thymic cellular interactome that maintains thymic homeostasis. Whether this decline in cellular interactions represents a cause or a consequence of the gradual accumulation of adipocytes within the thymus, and how it contributes to the process of thymic involution, remains to be determined. This observation underscores the utility of our system as a powerful tool to investigate biologically significant processes, such as the mechanisms underlying age-associated thymic degeneration. Notably, the identification of specific perturbed cellular interactions through this technology holds the potential to reveal therapeutic targets, thereby opening new avenues to prevent and/or reverse thymic involution.

R2C-4 (Minor-1): *The alignment of the right panel of Figure 6A is slightly off, which makes it difficult to understand the ligand and receptor interactions.*

Our response: We have now adjusted Figure 6A to improve alignment and reduced distance among arrows and graphs to make visualization easier.

Reviewer #3 – Comments (R3C):

R3C-1 (Minor-1): In Fig. 6A, it might have been informative to separate the DN_DP(P) group into DN and DP will clarify and provide more information regarding the interactions of each subset with the niche cells. For example, how does Notch signalling influence DN and DP cells differently, and which niche cells are responsible for these interactions at each stage?

Our response: We thank the Reviewer for this comment. We have tried to separate the DN_DP(P) group into DN and DP, but unfortunately the two clusters do not split up well. We included here plots showing sub-clustering of the DN_DP(P) cluster (Letter Figure 1). This cluster separated into two subclusters (0, in red, and 1, in blue) (Letter Figure 1, left panel). But these clusters do not perfectly correspond to DN and DP cells as some DPs appear in both clusters (Letter Figure 1, right panel provides markers). Thus cluster 1 is not just composed by DN cells. Perhaps the lower number of DN cells may be precluding their efficient separation.

Letter Figure 1.

R3C-2 (Minor-2): In Fig. 2B, it seems that the gating of mCherry-High cells may be a bit too generous. Some mCherry-Low cells are also included and could affect the downstream analysis.

Our response: We thank the reviewer for highlighting this point and have carefully reviewed the gates in Figure 2B. Now the background at 0 hour is down to 0.08%. Graphs were accordingly updated in Figure 2B. With the updated data, mCherry signal induction plateaued by 39-hours, which it is basically similar to previous analyses. This was included in line 180-182 as follows

Lines 180-182: “Co-culture of sender and receiver MEFs over time increased both mCherry MFI levels and the number of mCherryHIGH cells, with MFI reaching a plateau by 39 hours post co-culture (Fig. 2B).”

For more refined induction timing we have now included Figure S3B, where we demonstrated that three repetitive intermittent interactions of 30-minutes enables induction of a detectable mCherry signal as outlined in our response to Reviewer #1 point 2 (R1C-2).

R3C-3 (Minor-3): *In Fig.3E, the expression of mCherry rises beautifully in receiver cells when they interact with the senders. However, their mGFP level also rises, which is a bit confusing and the authors may want to address this concern.*

Our response: We thank the Reviewer for this comment. In Figure 3C, the Reviewer can find the distribution of sender cells in *CD4-Cre* mice. Most of senders are SP4s and DPs, but there are other sender cells, including SP8 and B cells and CD11c+ cells, hence the presence of GFP+ cells in the flow plots. They are included in Supplemental Table 2 and now detailed in the Figure 3C and its legend as follows:

Line 1360: “1.9% were B220+ cells and 0.6% CD11c+ cells. All percentages can be found in Supplemental Table 2.”.

R3C-4 (Editing/clarifications-1): *In Line 145, the authors could better define white blood cells by explaining the previous gating approach in their flow cytometric analysis? Was it only gated on FSC and SSC?*

Our response: The gating strategy for white blood cells is now included in Figure SF2Avi and was based on FSC and SSC, exactly as the Reviewer indicated.

R3C-5 (Editing/clarifications-2): *In Line 371-372, I would suggest using the following expression: “Thymuses from 20-weeks old.....served as controls for assessing the enrichment in mCherry+ cells....”*

Our response: We have amended the text as suggested:

Lines 420-421: “Thymuses from 20-weeks old CRE-negative mice served as controls for assessing the enrichment in mCherry+ cells in CD4-YY thymuses”.

R3C-6 (Editing/clarifications-3): *In the future, as part follow up work, it would be interesting to see comparisons between aged groups that are farther apart — for example, 1yr old vs. 8wks old mice.*

Our response: We thank the Reviewer for this encouraging suggestion for future work.

R3C-7 (Editing/clarifications-4): *In Line 422-423, I would suggest using the following expression: “Several components of the.....produced TGFb, which is crucial for SP4 differentiation to Tregs.”*

Our response: We have amended the text as suggested:

Lines 471-472: “Several components of the SP4-interactome, including B cells, mFbs, $\gamma\delta$ T17 and NKs, produced TGF β , which is crucial for SP4 differentiation to Tregs”

R3C-8 (Editing/clarifications-5): *In Line 423-426: “Pro-survival signalling as those triggered by...mediated by DCs and B cells...”*

Our response: We apologise for this typo and have amended the text as indicated.

Line 473: “triggered”.

Sánchez-Lanzas et al. response to Reviewers:

We thank the Reviewers for their very constructive and positive comments, which have helped us improve our manuscript. All changes are highlighted as “tracked changes” in the main manuscript word document, and our specific responses are detailed below, point-by-point.

REVIEWERS' COMMENTS

Reviewer #1:

The authors have carried out a thorough experimental and text revision that clearly and concisely addresses previous comments. In particular, new data in Figure S3B showing that 3-6 sequential transient interactions of 30' each is sufficient to induce mCherry expression in receiver cells demonstrates that the interactions recorded in the thymus microenvironment have the potential to reflect cognate interactions inducing positive or negative selection. These data support the relevance of the YinYang system to studying key cellular interactions driving selection events in situ. Moreover, the revised gating schemes in S2A and improve the accuracy of the cell types quantified as niche/receiver cells. Altogether, the authors have generated and carefully characterized an elegant system for studying cellular interactions during thymocyte differentiation and selection. I congratulate them on this important new model for the field and anticipate that many new insights will come from it.

Our response: We are delighted that the Reviewer appreciated our efforts and the new experimental data that we provided to support the YinYang system to study cellular interactions driving selection events.

There are two revisions needed (no experiments needed) prior to publishing to improve accuracy of data and interpretations presented:

RIC-1: *It is greatly appreciated that the authors have added flow cytometry gating for DP subsets to delineate which are senders, and it's notable that in the CD4-YY system, mainly post-positive selection DP subsets are senders (Figure 3Cii). However, the DP gating scheme needs to be revised because the non-sequential scheme in S2v results in overlap between subsets and thus, inaccurate quantification for Figure 3Cii.*

The gated “TCRab^{high} CD69+CD5+CCR7+ CCR4-“ DP cells are co-receptor reversing DPs, although they are currently labeled as “CCR4- post positive selection DPs”. These cells will overlap with the population currently labeled as “co-receptor reversal DP” because of the lack of sequential gating (both subsets are DP TCRab^{high}CCR7+ cells, as gated). I suggest not using additional restrictions on the CD4+CD8+ DP gate when identifying co-receptor reversing DPs, as these cells are in flux from CD4SP semi-mature cells and move through different levels of CD4 and CD8 as they fully differentiate to become CD8SPs. The co-receptor reversing DPs can be distinguished by expression of CD69 and high levels of CD3 or TCRab.

Also, In S2v, you've currently gated out most of the early post-positive selection DPs, as these are the TCRab low cells between your gates in the DAPI vs TCRab flow cytometry plot on the top row: to gate on these early post-positive selection DPs, you can gate CD69+ CD3 lo, or CD69+ TCRab low, cells within the DP gate, and you'll find that the majority (80-90%) express CCR4 but not CCR7. It's great that you have both of these chemokine receptor markers in your stain already.

To simplify and more accurately gate on non-overlapping DP subsets, I suggest drawing three gates to subdivide DPs, using a flow cytometry plot of CD69 vs either CD3 or TCRab: 1) CD69- and CD3lo or TCRab^lo DPs are the pre-selection cells (they'll express neither CCR4 nor CCR7); 2) CD69+ CD3lo or TCRab^lo DPs are early post-selection DPs (which are almost all CCR4+ CCR7-); 3) : CD69+ CD3hi or TCRab^hi are co-receptor reversing cells—this is a distinct population with good CD3 and CD69 resolution— (many of these express both CCR4 and CCR7, but there are some that express only one of these CCRs). With these 3 gates, you can easily distinguish three stages of DPs (pre-selection, post-selection, and co-receptor reversing) without any redundancy between them for accurate quantification.

Our response: We thank the Reviewer for these very insightful comments on the best gating strategies for DPs. We have now redefined the DP populations as suggested by the Reviewer (see below) and have updated the gating strategies in Figure SF2Av and in the Methods section (“Thymus analysis” lines 863-865) and distribution of sender populations in Figure 4Cii.

Pre-positive selection DPs: CD19⁻ CD11b⁻ NK1.1⁻ TER119⁻ CD4⁺CD8⁺ CD69⁻TCRαβ^{LOW}

Early post-positive selection DPs: CD19⁻ CD11b⁻ NK1.1⁻ TER119⁻ CD4⁺CD8⁺ CD69⁺TCRαβ^{LOW}

Co-receptor reversing DPs: CD19⁻ CD11b⁻ NK1.1⁻ TER119⁻ CD4⁺CD8⁺ CD69⁺TCRαβ^{HIGH}

The text describing these results has also been updated accordingly:

Lines 294-297: “Further analysis on the identity of DP senders revealed that 35% of them were pre-positive selection DPs (CD69⁻TCRαβ^{LOW}), 57% were co-receptor reversing DPs (CD69⁺TCRαβ^{HIGH}) (Fig. 4Cii). Early post-positive selection DP (CD69⁺TCRαβ^{LOW}) senders constituted a minor population (0.6%) (Fig. 4Cii).”

Lines 300-303: “However, given that 30% of the senders were DPs in CD4-YY thymuses, ~30% of the detected interactions would be driven by DPs (mainly co-receptor reversing DPs). Likewise, a minor proportion of the identified interactions would be triggered by SP8s (~7%).”

RIC-2: The previous Major Concern 4 was raised because DPs, CD4SPs, and CD8SPs were identified as receivers (of DP and CD4SP interactions) and their mCherry expression was reduced after MHC-II blockade, despite the fact that these receiver subsets don't express MHCII in mice. The authors have now added a thoughtful discussion (Lines 283-315) suggesting that prolonged MHCII blockade could result in secondary consequences to the thymus environment that reduce non-cognate interactions of with these thymocyte receiver cell types. However, they have retained the possibility that reduced mCherry in these subsets could indicate a reduction in MHCII-mediated thymocyte-thymocyte interactions, stating “Furthermore, thymocyte–thymocyte interactions have been shown to be important for “efficient positive selection and maturation of SP4 cells” [PMID: 16226504, PMID: 11292248, PMID: 21478404, PMID: 23264931, PMID: 20038602, PMID: 23776174], further supporting the biological significance of the interactions identified in this study.” This sentence should be removed, as the cited studies do not support that wild-type

thymocyte-thymocyte interactions in mice induce positive selection on MHCII. Choi EY et al., Immunity 23, 387-396 (2005) PMID: 16226504, and the other cited papers from this research group, use an LCK promoter-driven Ciita transgene to ectopically express MHCII on thymocytes. Mouse thymocytes do not otherwise express MHCII. Through ectopic MHCII expression, the authors demonstrate that mouse thymocytes CAN induce positive selection. If you look at the bone marrow chimeras in Figure 3 of PMID: 16226504, you'll see that WT BM transferred into Ciita-/- recipients does not result in positive selection of CD4SP cells, demonstrating there is no natural thymocyte-thymocyte mediated positive selection on MHCII. Because mouse thymocytes don't naturally express MHCII, in the current study, the reduced interactions of CD4SP with thymocyte subsets strongly implicates a secondary effect of MHCII blockade on the thymus environment, as opposed to a direct effect of interfering with cognate interactions. The idea, however, that thymocyte-thymocyte interactions induce agonist selection of innate thymocyte subsets is well supported by the literature (but this isn't driven by MHCII-mediated interactions naturally, although the ciita transgenic model led to important insights about innate lymphocyte selection). Also, human thymocytes express MHCII, so they may contribute to positive selection of CD4SP, but that it is not relevant for results in this mouse model.

Our response: We thank the Reviewer for these really insightful comments. As suggested we have removed the indicated sentence from Line 438:

~~“Furthermore, thymocyte-thymocyte interactions have been shown to be important for “efficient positive selection and maturation of SP4 cells” [PMID: 16226504, PMID: 11292248, PMID: 21478404, PMID: 23264931, PMID: 20038602, PMID: 23776174], further supporting the biological significance of the interactions identified in this study.”~~

Reviewer #2:

The authors have addressed my previous concerns, and I have no further comments. The technologies reported in this manuscript will have a broad impact for mapping cellular interactomes in vivo, and the reported data provide new insights into thymus biology.

Our response: We are delighted on Reviewer's comments and thank them for their previous suggestions.

Reviewer #3:

The authors have fully and carefully addressed all my initial concerns, and should be congratulated on their exciting new mouse model system and for the insights derived from their analysis.

Our response: We are delighted on Reviewer's comments and thank them for their previous suggestions.